# Hospital length of stay: A cross-specialty analysis and Beta-geometric model

**Nassim Dehouche[1], Sorawit Viravan[2]\*, Ubolrat Santawat[2], Nungruethai Torsuwan[2], Sakuna Taijan[2], Atthakorn Intharakosum[2], Yongyut Sirivatanauksorn[2]**

**1** Business Administration Division, Mahidol University International College, Salaya, Thailand, **2** Faculty of Medicine Siriraj Hospital, Mahidol University, Bangkok, Thailand

\* Sorawit.vir@mahidol.edu

## Abstract

### Background

The typical hospital Length of Stay (LOS) distribution is known to be right-skewed, to vary considerably across Diagnosis Related Groups (DRGs), and to contain markedly high values, in significant proportions. These very long stays are often considered outliers, and thin-tailed statistical distributions are assumed. However, resource consumption and planning occur at the level of medical specialty departments covering multiple DRGs, and when considered at this decision-making scale, extreme LOS values represent a significant component of the distribution of LOS (the right tail) that determines many of its statistical properties.

### Objective

To build actionable statistical models of LOS for resource planning at the level of healthcare units.

### Methods

Through a study of 46, 364 electronic health records over four medical specialty departments (Pediatrics, Obstetrics/Gynecology, Surgery, and Rehabilitation Medicine) in the largest hospital in Thailand (*Siriraj Hospital in Bangkok*), we show that the distribution of LOS exhibits a tail behavior that is consistent with a subexponential distribution. We analyze some empirical properties of such a distribution that are of relevance to cost and resource planning, notably the concentration of resource consumption among a minority of admissions/patients, an increasing residual LOS, where the longer a patient has been admitted, the longer they would be expected to remain admitted, and a slow convergence of the Law of Large Numbers, making empirical estimates of moments (e.g. mean, variance) unreliable.

### Results

We propose a novel Beta-Geometric model that shows a good fit with observed data and reproduces these empirical properties of LOS. Finally, we use our findings to make practical recommendations regarding the pricing and management of LOS.

**Data Availability Statement:** All relevant data are within the manuscript and its Supporting information files.

**Funding:** The author(s) received no specific funding for this work.

**Competing interests:** The authors have declared that no competing interests exist.

# Background

## Introduction

In a global healthcare sector that is gradually but steadily shifting from a fee-for-service to value-based care agreements, Length of Stay (LOS) is a useful indicator of resource utilization and cost-efficiency, and has been likened to a currency for healthcare decision-making [1, 2]. Because expenses are largely fixed in the first few days following admission, the marginal value of a bed is determined by the alternate use that the healthcare provider would have made of it (i.e. its opportunity cost) [2]. This value may be neglectable for low-utilization healthcare units. However, for high-demand hospitals operating close to capacity, the opportunity cost of long LOS can be significant. This is the case for Siriraj Hospital, the oldest and largest hospital in Thailand. Siriraj being an affordable, yet very reputed healthcare provider in the country, its specialty departments experience very high demand, often resulting in long waiting lists for admissions that can, in some cases, take several months.

In such circumstances, capacity planning has been recognized as a critical decision for the effective management of the hospital's resources, and because key resources are shared across different Diagnosis Related Groups (DRG), understanding the statistical properties of LOS and constructing accurate statistical models of it, at the relevant scale of a healthcare unit, constitutes as a crucial input for capacity planning.

In this didactic statistical study, 46, 364 electronic health records over four medical specialty departments were analyzed, with a focus on the tail properties of the distribution of LOS, and their managerial consequences. The remainder of this paper is organized as follows. In Section Related Work, we conduct a detailed review of the related literature and classify extant studies based on their sample sizes, underlying statistical models, scope, and treatment of outliers. Section Data Description and Basic Statistical Properties describes our data and presents basic descriptive statistics for LOS in the four considered specialty departments. Section Materials and Methods defines the theoretical framework of this paper, which consists in various statistical and graphical tests from Extreme Value Theory, as well as the Beta-Geometric model we propose for LOS. Sections Results and Discussion and Modeling Length of Stay present our results and Section Managerial Implications further elucidates their implications for profit margins and resource management. Finally, Section Conclusions and Limitations concludes this paper with general remarks concerning our statistical analysis and its limitations.

## Scope of decision-making

Hospital capacity planning [3], an essential component of healthcare management requires accurate statistical models of LOS. Recent events such as the Covid-19 pandemic have further highlighted the importance of this fact [4].

Approaches to hospital capacity planning are diverse, with statistical modeling to predict future demand being a common strategy. Section Related Work proposes taxonomy of these models. Another widespread approach involves using simulation models to test different capacity scenarios. Such models often incorporate the statistical models of LOS to fine-tune predictions.

For instance, Pearson et al. [5] used Monte Carlo simulations in conjunction with chosen random variables for LOS to estimate the need for Intensive Care Beds across several DRGs. Devapriya et al. [6] designed StratBAM, a capacity planning software that takes into account the distribution of care length at each care unit level, highlighting the centrality of choosing the right model for LOS.

Even though LOS models based on DRGs have received significant attention due to their utility in clinical decision-making (as discussed in Section Scale of modeling, recent developments have called for "geekier" models for capacity requirements [7]. These models see the hospital as an interdependent system [3], further underscoring the May 24, 2023 2/28 significance of choosing the correct random variable for LOS modeling in capacity planning. As suggested in [8], although DRGs are useful for pricing calculations, they offer limited guidance on the mix of resources needed.

Thus, it is this specific aspect—the decision of the statistical model to represent LOS—that our present paper primarily addresses. However, it is crucial to remember that these statistical models form only one piece of the complex hospital capacity planning puzzle. Other factors, including clinical variables and qualitative inputs, also contribute significantly to a comprehensive capacity planning decision support system. Thus, while our study underscores the importance of making the correct decision about the statistical model for LOS, it also appreciates the broader context within which this decision is made.

## Related work

A distribution is said to be *heavy-tailed* if its survival function $S(t)$ satisfies $e^{\lambda \cdot t} S(t) \to +\infty$ as $t \to +\infty$ [9]. Its moments, including the mean and variance, can be infinite. A distribution is called thin or light-tailed otherwise. Though it exhibits the empirical properties of heavy-tailed distributions, Hospital Length of Stay (LOS) is often modelled with thin-tailed distributions for the purpose of least square regression. Indeed, the typical Length of Stay (LOS) distribution is known to be right-skewed, to vary considerably across Diagnosis Related Groups (DRG) [10], and to contain markedly high values, in significant proportions. These characteristics make the use of thin-tailed models and least-squares inference methods based on the Gauss-Markov theorem [11] hard to justify. Moreover, they make the calculation of averages less reliable and representative of the typical observation [12]. Extant works on the statistical modelling of LOS attempt to circumvent these difficulties, with the following solutions, reviewed in this section:

1. Using different models of LOS for different DRGs in a care unit.

2. Trimming or discarding outliers.

3. Using different models for short and long stays (i.e. mixtures of distributions).

4. Relying on heavy-tailed models.

Table 1 summarizes the characteristics of the main references on statistical models of LOS in the literature.

**Scale of modeling.**   According to Ickowicz et al. [16], *Diagnosis Related Groups* (DRG) have been partly created to have a form of homogeneity in resource consumption of services and costs closely related to LOS. Models of LOS at the DRG level can be valuable for clinical decision-making (e.g. determining which individual surgeries can be postponed [21]), and understanding the specific determinants of LOS, seen as a proxy for the severity of a patient's condition, via its association with independent variables of a demographic, lifestyle, or medical nature is often favored and is used to predict an individual's expected LOS.

However, this approach results in smaller samples by design, which can make very high LOS values indeed appear as neglectable outliers. At the scale of a healthcare unit treating different DRG, these "outliers" have been empirically found to utilize a disproportionate and increasing share of hospital resources and available beds [22].

**Table 1. Characteristics of extant statistical models of LOS.**

| Reference | Dataset size (N) | Model | By DRG | Outliers |
|---|---|---|---|---|
| [2] | 1901 (in 2 hospitals) | Markov chains | No | No |
| [9] | 137 + 469 | Mixture of Exponential and heavy-tailed | Yes (1 DRG) | No |
| [10] | 560 | Mixture of Gaussian | Yes | Trimmed |
| [12] | 4, 758, 347 (in 5 countries) | Log-normal, Weibull, Gamma | Yes | No |
| [13] | 3, 472 | Gaussian | No | Separated |
| [14] | 9, 040 | Gaussian assumptions (use of Z-scores) | No | Separated |
| [15] | 4, 570 | Gaussian assumptions (use of interquartile range rule) | No | Separated |
| [16] | Simulated | Mixture of continuous and discrete | Yes | No |
| [17] | 340 | Log-normal | No (ICU) | No |
| [18] | Unknown (in 1 hospital) | Mixture of thin and heavy-tailed | Yes (5 DRG) | No |
| [19] | 53, 965 | Gaussian assumptions (use of Z-scores) | No | Separated |
| [20] | 101, 766 (in 136 hospitals) | Heavy-tailed compounds | No (diabetes) | No |
| This paper | 46, 364 (in 1 hospital) | Beta-Geometric | No | No |

Moreover, there exist more than 467 different DRG, and combining even excellent LOS models for different DRG is non-trivial and does not necessarily yield valid models for resource management at the level of a care unit covering multiple DRGs, due to the complex non-linear nature of healthcare systems [23]. Thus, this approach transfers the modeling difficulty towards re-aggregating results (e.g. averages or medians per DRG, linear regression forecasts) at a scale that can inform decision-making. A task as simple as estimating the variance, coefficient of variation, and kurtosis of LOS at the level of a care unit, based on the distribution of LOS by DRG is a non-trivial problem, even when all distributions are known to be Gaussian [24], which is far from being the case for LOS. Besides, the non-transitivity of (Pearson's) correlation [25] makes associative inference on the parts not necessarily scalable to the whole.

Ickowicz et al. [16] note that the "heterogeneity of LOS poses a problem for statistical analysis, limiting the use of inference techniques based on normality assumptions since a large number of DRGs must be analyzed routinely, automatic procedures are needed for conveniently treating skewness" and add that "the main issue is that the assumption of heterogeneous sub-populations would be more appropriate than single DRG populations". Their proposed solution is the use a mixture of probability distributions. The same point has been made by Atienza et al. [18], who conclude that "the assumption of heterogeneous sub-populations would be more appropriate than single DRG populations".

**Treatment of outliers.** Grubbs [26] defines an *outlier* as an observation that "appears to deviate markedly from other members of the sample in which it occurs".

For the positive random variable that is LOS, outliers are conventionally considered to be admissions or patients with a remarkably high LOS. Under Gaussian assumptions, empirical Length of Stay (LOS) datasets are commonly described as containing outliers.

Perhaps as a side-effect of the small samples sizes typically considered in the above studies, the notion that high-LOS admissions are outliers appears founded. Indeed, in their application using empirical data, [10] consider $N = 560$ observations clustered within 21 hospitals, with the numbers of patients ranging from 3 to 196 per hospital.

Trimming outliers, that is separating data into normal and high-LOS, is commonly performed. Thus, the question of how to define thresholds for what is considered long or short LOS is important from an operational and financial point of view and at the center of many studies.

The model considered in [13] is typical of such a modeling approach, which the authors remark in [27] is not of their own design but a more than 30 years old standard risk model of the Society of Thoracic Surgeons [28].

In [29], the threshold for what is considered long LOS is set at seven days or as the top 2% of LOS values which has been qualified as "somewhat arbitrary but has been applied by others in the analysis of administrative data" [10, 30]. "The objective of trimming coupled with transformation is to minimize the effects of extreme outliers and to attain the normality assumption on the LOS distribution" [10], as well as "to minimize the effects of extreme outliers and to avoid analytical problems" [29]. Leung et al. suggest trimming LOS observations at the threshold of seven days.

The 68% − 95% − 99.7% rule is also commonly used [12], and outliers are defined as observations that lay at a distance of more than three standard deviations from the mean, thus assuming a Gaussian distribution of LOS.

However, what are considered outliers for individual DRG may be statistically very significant in the study of LOS at the larger scale of a healthcare unit. Consequently, trimming/discarding outliers may discard important statistical information about the tail properties of LOS at this scale of decision-making. Lee et al. [10] transparently note that the use of trimming methods for outliers limits the usefulness of the models, they add "if the goal of the analysis is plan or hospital comparison, uniform trimming of LOS across all hospitals might be inappropriate in the contexts of quality improvement and performance assessment".

High LOS admissions are known to have a prominent financial impact. This point is stated by [15], which proposes a method that notably serves at "detecting outliers".

Lee et al. [10] recommend relying on the median rather than mean estimates in the analysis of LOS, because the latter approach is more "robust to high-LOS outliers". However, the robustness of the method is, circularly, gauged on Gaussian simulated data with Bernoulli noise.

Gardiner et al. [9] note that "in modeling hospital Length of Stay (LOS) and inpatient cost, extreme values in the data are likely and should not be regarded as outliers for deletion or downweighted in analyses". The heavier the tail of a distribution, the more statistical information it contains relative to the body [31].

Using large datasets covering four medical specialties, the present article will show that the observations considered outliers in thin-tailed models are in fact too numerous to be outliers, and they have a significant impact on resource consumption and revenue.

**Use of mixture distributions.** A *mixture distribution* is the probability distribution of a random variable that is sampled from two or more different probability distribution functions (PDF) [24]. The PDF of the mixture random variable is often a weighted sum of the PDFs of the mixed random variables. We distinguish *mixture distributions* from *compound distributions*, the latter being distributions whose parameters are themselves random variables [32].

The problem with mixture probabilities may be overfitting, especially with the small sample sizes that are typically considered. As noted in [27], this is an important pitfall in modeling LOS.

The model proposed by Atienza et al. [18] divides patients by DRG and relies on a mixture of thin and heavy-tailed distributions (Gamma, Weibull, Log-normal). However, its application was limited to simulated data (128 samples of size 100 and 100 samples of size 500).

Gardiner et al. [9] consider a first sample of $N = 137$ patients who underwent bone marrow surgery and a second sample of $N = 469$ patients in Psychiatry. In the latter sample, they interestingly report LOS varying from 1 to 24, 028 days, with a mean of 3, 712.4 days and a median of 1, 134 days. The authors recommend a mixture of exponential and heavy-tailed distributions (Pareto, Generalized Pareto).

Lee et al. [10] consider one DRG in Obstetrics and Gynaecology ("Cesarian delivery with severe complicating diagnosis"). They consider an empirical dataset of $N$ = 560 patients. Outliers are trimmed. They use a mixture of (Gaussian) distributions.

Ickowicz et al. [16] consider a model for "short stays" and a different model for "long stays", resulting in a mixture of continuous (Normal and Log-normal) and discrete (Poisson, Binomial, Negative Binomial) random distributions. Their sample size is unknown but it is mentioned that it is similar to that of [18], which consists in simulated data.

Rady et al. [17] consider $N$ = 340 patients of an Intensive Care Unit. No division by DRG is performed, which is justified by the relatively low range of LOS in this sample (the minimum observed LOS was one day, and the maximum was 60 days).

**Treatment of tails.** A recent debate [27] concerning the statistical representation of LOS is a thoracic surgery department [13] highlighted the prevalence of thin-tailed statistical modeled in this context.

Though no assumptions are explicitly made regarding the distribution of LOS, [13] use linear regression on averages and medians, which requires a Gaussian underlying distribution. This point was criticized by [27] which observes that "ordinary least squares regression does not adequately accommodate large LOS values" and recommends "normality-improving data transformations", which typically consist in Log-transformations. However, [2] point out that the weakness of this approach is that "Log-LoS" does not have an intuitive meaning and is therefore not useful for policy making. Moreover, the retransformation of the regression results from log results through exponentiation is complicated by heteroscedasticity (i.e. the fact that the variability of LOS is unequal across the range of values of the independent variables used to predict it), and produces "very imprecise estimates if the log-scale error is heavy-tailed". In fact, single-point forecasts are not theoretically justified for heavy-tailed variables with high standard deviation [33]. Moreover, the mean and other moments may be infinite or require a very large number of observations to be estimated, due to the slow convergence of the Law of Large Numbers for variable of this type. This point has also been highlighted by [9].

A more robust approach consists in modeling the relevant properties for capacity planning (e.g. through fitted distributions) rather than predicting punctual values.

Marazzi et al. [12] assessed the adequacy of three conventional parametric models, Log-normal, Weibull and Gamma, for describing the LOS distribution. But, as Lee et al. [10] point out, none of them seemed to fit satisfactorily in a wide variety of samples.

Harini [20] use data from a public repository covering 136 hospitals and $N$ = 101, 766 diabetes patients. No division by DRG is performed beyond that. They recommend heavy-tailed distributions (Beta-Cauchy, Gamma-Pareto, Gamma-Exponential-Cauchy). It should be noted that these are compound distributions, not mixtures of distributions. Moreover [12] suggest two approaches to respond to the skewed nature of the distribution of LOS and building "outliers resistant (robust) methods"; using other transformations than the Log-normal or using other types of distribution. The authors pursue the latter approach and test three heavy tailed distributions (Log-normal, Weibull, Gamma), which are found to adequately fit the distribution of LOS, over 3279 samples totaling approximately 5 million stays in multiple European countries. Among these, the Log-normal model is found to fit the majority of samples.

Based on a dataset recording 1901 patients' LOS, Faddy et al. [2] develop an intuitive continuous Markov process which is found to provide a better fit than Gamma and Log-normal models. This model is based on the assumption that each day of admission, patients either progresses to another day of hospital stay, corresponding to increasing their LoS, or they are discharged (absorbing state). This defines a general class of probability distributions describing the random time that elapses before the absorbing state is reached. The Beta-Geometric model of LOS developed in the present article is based on the same intuition. Our model rests on the

two assumptions that once admitted a patient has a certain likelihood of discharge any subsequent day, and that different patients have different such likelihoods.

However, because of the discrete nature of LOS, we consider a discrete Beta-Geometric process. Thus, our model differs and complements the reviewed literature as follows:

- We consider large samples of empirical observations ($N = 46, 364$ over four medical specialty departments).

- The Beta-Geometric model we propose is discrete, right skewed and can fit thin to very heavy-tailed behavior, depending on the choice of parameters of the distribution.

- The object of our analysis are both the LOS per admission as well as the LOS per patient, highlighting the effect of multiple admissions/readmissions on the statistical properties of LOS.

- The scale of our model are care units corresponding to four medical specialties, but the model can be scaled up (e.g. to a hospital) or down (e.g. DRG) without loss of generality.

- We propose various novel graphical tools to complement LOS analysis. Notably the use of Mean Excess Plots to detect mixtures of probabilities in and thresholds.

- We highlight and model important properties for resource management, such as the behavior of the mean excess function, Maximum to Sum ratios and the effect of the concentration of LOS among a minority of admissions/patients on bed turnover and revenue.

## Data description and basic statistical properties

The dataset used in this study covers four medical specialty departments at Siriraj Hospital. These are Surgery, Obstetrics and Gynaecology (OB/GYN), Pediatrics, and Rehabilitation Medicine (Reh. Med.).

The following data fields were utilized:

- Admission Number: A unique identifier for each admission.

- Patient ID: A unique identifier for each patient.

- Admission Date.

- Discharge Date.

- Length of Stay: The difference in days between Discharge Date and Admission Date, the minimum value of which is 1 days. Consequently, random variables modeling LOS are truncated if having support on negative values, and shifted by one day, if having support on zero.

- Total Charge: The total billing expense to the payer, in Thai Baht (THB), resulting from an admission.

- Discharge Status: We distinguish positive discharge outcomes ("Complete recovery", "Improved", "Delivered", "D/C with mother") and negative ones ("Dead", "Not improved", "D/C separately"), when applicable to admissions in a medical specialty.

Throughout this paper, we distinguish between LOS per admission (number of days between admission and discharge for one admission, identified by an admission number) and LOS per patient (cumulated LOS for all admissions of a patient identified by a patient ID). These two levels of analysis notably allow us to study the effect of readmission and multiple admissions on the tails of the distributions of LOS. Realizations of both of these integer

**Table 2. Descriptive statistics for the LOS per admission and per patient in the four departments.**

| Statistic | Surgery | OB/GYN | Pediatrics | Reh. Med. |
|---|---|---|---|---|
| **N⁰ of Admissions** | 17949 | 19922 | 7499 | 994 |
| Mean | 6.5973 | 3.8533 | 8.2515 | 24.5824 |
| Median | 4 | 3 | 4 | 21 |
| Variance | 125.6517 | 11.0901 | 264.5326 | 187.2465 |
| Dispersion | 19.0459 | 2.8780 | 32.0587 | 7.6170 |
| Kurtosis | 139.0954 | 183.9783 | 47.8032 | 4.7843 |
| Skewness | 8.9926 | 9.4236 | 5.9014 | 1.9304 |
| Max | 304 | 119 | 272 | 87 |
| **N⁰ of Patients** | 14884 | 17483 | 5086 | 304 |
| Mean | 7.9558 | 4.3908 | 12.1663 | 80.3782 |
| Median | 4 | 4 | 4 | 66 |
| Variance | 215.4532 | 15.9265 | 715.3997 | 2613.087 |
| Dispersion | 27.0810 | 3.6271 | 58.8015 | 32.5098 |
| Kurtosis | 279.3174 | 134.4066 | 56.51553 | 6.428852 |
| Skewness | 12.18991 | 8.437034 | 6.163459 | 2.278371 |
| Max | 546 | 119 | 451 | 321 |

random variables, along with complete administrative and medical data are recorded for discharges that occurred between 13/11/2017 and 30/09/2018, the earliest recorded admission having occurred on 15/02/2017 and the latest on 29/09/2018.

Table 2 presents the sample sizes as well as basic descriptive statistics in each medical specialty department. We note that all distributions fail the Skewness-Kurtosis test for Truncated Gaussians [34]. Moreover, we note a markedly higher mean and median LOS per admission and per patient in Rehabilitation Medicine compared to the three other specialties, a fact that was also observed in [19]. These two points are related since the time-frame of data collection is the same for all specialty departments. Additionally, the Rehabilitation Medicine department has a smaller capacity. Further, this department shows the smallest ratio of patients-to-admissions, indicating frequent multiple admissions over the time-horizon of the study (10.5 months). Indeed its 994 total admissions are attributed to 304 patients (patients-to-admissions ratio of 30.52%), to be compared with the ratios of 87.75% in OB/GYN, 82.92% in Surgery, and 67.82% in Pediatrics. Moreover, the distribution of LOS in this medical specialty exhibits a significantly lower Kurtosis. Indeed, Kurtosis in the distribution of LOS in Surgery, OB/GYN and Pediatrics is extremely high and constitutes a first, important signal of the heavy-tailedness of the distribution of LOS in these departments. This allows us to safely reject thin-tailed distributions (Gaussian, Poisson, etc.) at this point for these three departments.

Among these three specialties, OB/GYN markedly differs by its lower Index of dispersion (i.e. variance to mean ratio). Moreover, Kurtosis significantly increases when aggregating LOS per patient, except for this medical specialty. Lastly, we note the remarkable stability of the Median LOS in Surgery, OB/GYN and Pediatrics, as a likely result of the adherence to Benchmarks in the management of LOS.

The distributions of LOS in the departments of Surgery, Pediatrics, and OB/GYN were found to exhibit similar statistical properties. Therefore, for the sake of simplicity, the body of this paper will only present our results for one of these three departments and contrast them with the statistical properties of LOS in Rehabilitation Medicine.

Results for the omitted departments can be found in S5 to S11 Figs.

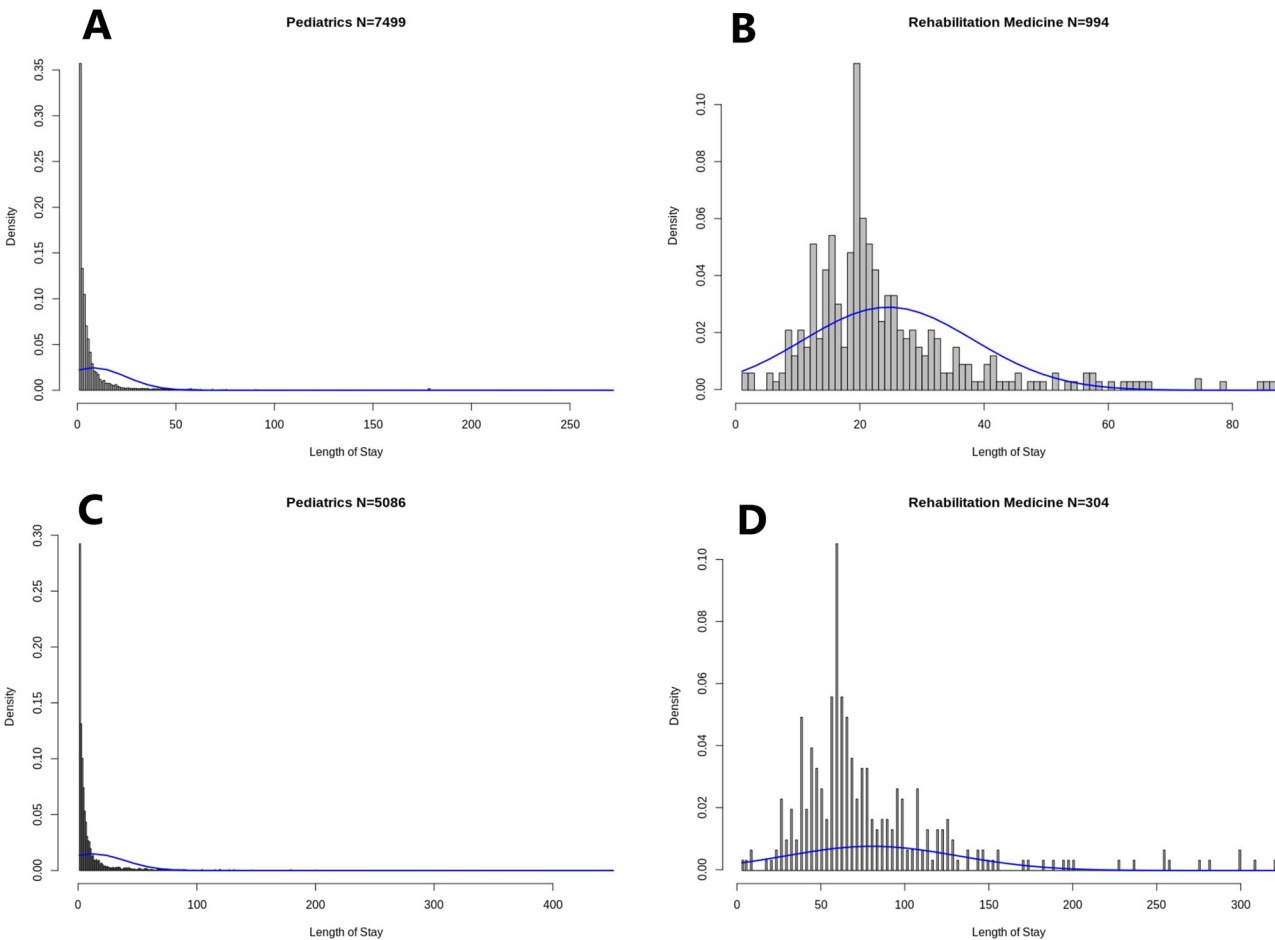

**Fig 1. Histogram of LOS.** (A) Histogram of LOS per admission in Pediatrics. (B) Histogram of LOS per admission in Rehabilitation Medicine. (C) Histogram of LOS per patient in Pediatrics. (D) Histogram of LOS per patient in Rehabilitation Medicine.

Histograms further confirm the singular nature of Rehabilitation Medicine relative to the three other specialty departments. Fig 1B presents the histogram of LOS, per admission, in Rehabilitation Medicine over which the maximum likelihood truncated Gaussian fit is super-imposed (blue curves). We observe a strikingly high frequency for the particular LOS value of 20 days, which for once fits the definition of an outlier as "an observation that differs significantly from other observations" [26]. Indeed, this value shows a frequency of 144 admissions, which represents 11.46% of the 994 admissions in Rehabilitation Medicine. This markedly differs from neighboring values of 19 days with a frequency of 48 admissions (4.8% of all admissions), and 21 days with a frequency of 60 admissions (6.0% of all admissions). A LOS of 20 days was previously found by Chatterjee et al. [35] to appear in remarkably similar relative proportions to 19 and 21 days (about twice as frequently) at the Skilled Nursing Facilities (SNF), where 220, 037 discharges were made on day 20, compared to 131, 558 and 121, 339, on days 19 and 21, respectively. Chatterjee et al. [35] explain this common anomaly by the fact that the first 20 days of rehabilitation at the SNF are covered in full, and patients start paying out-of-pocket from day 21, which motivates discharge on the 20th day of care based on a concern for a patient's ability to pay rather than their recovery status. Additionally, this arbitrary but

"round" value may more simply represent a psychological anchor and its frequency may be explained in an analogous manner to Benford's law for digits [36].

In the case of Siriraj hospital, this anomalous value is explained by a policy at the Rehabilitation Medicine department that aims at controlling the average LOS.

From a statistical perspective, the fact that some stays are artificially cut short at 20 days suggests that LOS would exhibit heavier tails without this manual restriction. Fig 1D, which represents the LOS per patient in the same specialty department shows a similar effect, with a cumulated LOS of 60 days being the overwhelmingly most frequent value (a frequency of 32 observations or 10.52% of the 304 patients) compared to the two neighboring values of 57 and 63 cumulated days (both observed 17 times or 5.59% of the 304 patients).

Though the distribution of LOS in Rehabilitation Medicine exhibits the thinnest right tail of all specialty departments, we can observe in Fig 1B, and even more so in Fig 1D, that it nevertheless does not fit a Gaussian distribution. Indeed extreme values (e.g. above 60 days for LOS per admission, and above 200 cumulated days for LOS per patient), which would be close to impossible to observe in a Gaussian, Poisson, or Geometric distribution of similar mean of variance, are empirically way too frequent. The inadequacy of thin-tailed models is, as expected from Kurtosis, exceedingly more pronounced for Pediatrics, Surgery, and OB/GYN, as can be seen in Fig 1A, S5A and S5B Fig representing the histograms of the respective distributions of LOS per admission in these departments, as well as Fig 1C, S5C and S5D Fig, representing the histograms of their respective LOS per patient, over which the best-fitting truncated Gauss-ian curves are superimposed.

## Materials and methods

The present study was approved (Certificate of Approval No. Si 032/2021) by the Institutional Review Board of Mahidol University, which waived patients consent. All data provided to the first author for this study were de-identified.

Besides the more common Gini Index and Lorenz curves presented hereafter, this study utilizes some statistical methods from the Extreme Value Theory toolset whose usage is not widespread in the medical literature (though they constitute staples of the mathematics of insurance). This section provides a brief description of these methods. Additional methodological details and illustrations of these methods can be found in S1 Fig. Moreover, for the sake of conciseness, the body of this manuscript only presents results for the two specialty departments presenting the most notable properties, among the four considered specialty departments. Results corresponding to the two omitted departments are presented in S5 to S11 Figs.

### Gini index and Lorenz curves

The Gini index and Lorenz curves are classical statistical indicators of inequality. The Lorenz curve [37] is a representation of the CDF of a random variable showing the proportion of its total value that is concentrated in the bottom $x$% of observations. It is often used to represent income distribution, where it shows the percentage $y$% of the total wealth owned by the bottom $x$% of households. Inequality is represented as the area separating the Lorenz curve from the $y = x$ line, corresponding to the Lorenz curve of the Dirac delta distribution, i.e. perfect equality.

The value of the Gini index [37] represents the percentage of the area between the line of perfect equality of distribution and the observed Lorenz curve. Possible values range from 0 to 1, higher values indicating less equal distributions. The "80–20" Pareto Principle, for instance, is indicated by a value of the Gini index of approximately 0.76 [38].

## Maximum to Sum ratios

Heavy-tailed distributions raise the question of the convergence of empirical moments (the first and second moments being the mean and variance) [39]. Given an order $p \in \{1, 2, 3, 4, \ldots\}$, the convergence of the ratio of the maximum to the sum of exponent $p$ is indicative of the existence of the moment of order $p$, and if so of the speed of convergence of the empirical moments of order $p$ to its true value. Formally, given a sample of $n$ observations $\{x_1, \ldots, x_n\}$ of a positive random variable $X$, let $M(n, p) = Max\{x_1^p, \ldots, x_n^p\}$ be the maximum of order $p$ and $S(n, p) = \sum_{i=1}^{n} x_i^p$, the sum of order $p$. We have the following result [39, 40]:

$$E(X^p) < +\infty \Leftrightarrow \lim_{n \to +\infty} \frac{M(n, p)}{S(n, p)} = 0$$

Based on the previous equivalence, Maximum to Sum plots [41, 42] represent the ratio of the maximum to sum of order $p$ as a function of the number of data points for different values of $p$ and indicate a convergence of the moments of order $p$ to a finite value if and only if the ratio converges to zero.

## Mean excess functions

The excess distribution over a threshold $a$ for a random variable (such as a duration, e.g. LOS), with support in $D(X)$, is defined [43, 44] as $F_a(x) = P(X - a \le x | X > a)$, $a \in D(X)$. Intuitively, its complement $1 - F_a(x)$ measures the likelihood of $X$ exceeding $a + x$, given that $X$ has exceeded $a$. For instance, if $X$ measures LOS, $1 - F_a(x)$ is the likelihood of a patient staying $x$ more days, given that they have been admitted for $a$ days so far. The Mean Excess function, also known as the Mean Residual Life function, is the expectation of this distribution for random variable of finite expectations and is defined as $ME(a) = E(X - a | X > a)$, $a \in D(X)$. If $X$ measures LOS, this would be the expected remaining LOS of a patient, given that they have been admitted for $a$ days so far.

This excess distribution and mean are the foundations for peaks over threshold (POT) modeling [44] which fits distributions to data on excesses and has wide applications notably in risk management, actuarial science, project management, and survival analysis. Moreover, they define three classes of random variables whose life-expectancy exhibits crucially different statistical behaviors:

- A decreasing mean excess function is characteristic of thin-tailed random variables with memory. If the variable measures the duration of a certain state, the longer an object has been in that state, the lower the expected remaining duration.

- A constant mean excess function is characteristic of *memorylessness* [45]. Exponential random variables and their discrete analogues, Geometric random variables, are known to exhibit this property.

- An increasing mean excess function is characteristic of scalable heavy-tailed random variables and corresponds to the *Lindy Effect* [46] where "the longer you wait, the longer you will be expected to wait". For instance, the longer a book has been in print or a project has been running late, the longer their expected remaining duration in a state of print or tardiness respectively, and any additional period in those states increases the expected remaining duration. Lindy things "age in reverse" [47].

## The subexponential class of distributions

Let $X = X_1, \ldots, X_n$ be a sequence of positive independent and identically distributed random variables with cumulative distribution function (CDF) $F$. Following [48–50], we consider that $X$ belongs to the subexponential class if it satisfies the following property:

$$\lim_{x \to +\infty} \frac{1 - F^{(2)}(x)}{1 - F(x)} = 2$$

Where $F^{(2)}$ is the CDF of $X_i + X_j$, $i, j \in \{1, \ldots, n\}$, the sum of two independent copies of $X$. Practically, this property implies that likelihood of two independent observations of $X$ (e.g. two patients' LOS) exceeding a high threshold $x$ is twice the likelihood of either one of them exceeding $x$. Thus, for only two observations, the value of the sum, if high, is dominated by an individual observation and the other one contributes negligibly. This property can be extended to $n$ observations [48–50], where the property $\lim_{x \to +\infty} \frac{1 - F^{(n)}(x)}{1 - F(x)} = n$ also characterizes the subexponential class. Therefore the sum of $n$ observations would be dominated by extreme values, which makes aggregate indicators based on the sum (e.g. the mean) less indicative of a typical value and more sensitive to extreme values. In distributions verifying this property, extreme observations can disproportionately impact and determines sums or aggregates, an example of which being the wealth of a group of people in which the wealthiest person in the country is included [46]. The total or average wealth of the group would be overwhelmingly determined by the wealth of that person. This property is also known as the *catastrophe principle* [50]. For subexponential distributions, the simplest explanation for a large sum and thus mean is that one large observation happened, not that a collection of many slightly larger than expected events conspired together to make the sum large. This property runs completely contrary to what happens under model thin-tailed distributions considered to model length of stay (Gaussian, Poisson), but is present in Log-normal models and the Beta-Geometric model we propose herein.

Another important consequence of this property is the inapplicability of the Gauss-Markov theorem [48] and thus of linear least-squares regression methods. However, maximum likelihood estimation methods [51] are applicable in this context [48].

## The Beta-Geometric distribution

Our proposed model for hospital Length of Stay is based on the following two assumptions, in which $X$ is a random variable representing its value for an individual admission or patient:

1. Once admitted, a patient has a constant probability $p$ of being discharged any subsequent day. Given $p$, this is equivalent to assuming that $X$ follows a (shifted, i.e. starting from 1 instead of 0) Geometric distribution, with probability density function (PDF)
$P(X = x|p) = p \cdot (1 - p)^{x-1}, x \in \{1, 2, 3, \ldots\}$

2. Patients within a medical unit, department, or hospital exhibit different probabilities of discharge $p$ depending for instance on their diagnosis related group, individual health condition, type of care, hospital policy, and other factors. In other words, $p$ itself is a random variable. The Beta distribution [52] is conventionally used to model the variations of a probability in a population. It is characterized by two positive real parameters $\alpha$ and $\beta$ and gives the following PDF for $p$: $f(p|\alpha, \beta) = \frac{p^{\alpha-1} \cdot (1-p)^{\beta-1}}{B(\alpha,\beta)}$, $p \in [0, 1]$, where $B()$ is the Beta function given by $B(\alpha, \beta) = \frac{\Gamma(\alpha+\beta)}{\Gamma(\alpha) \cdot \Gamma(\beta)}$ and $\Gamma()$ is the Gamma function given by $\Gamma(\alpha) = \int_0^{+\infty} x^{\alpha-1} \cdot e^{-x} dx$.

Assuming a count that starts at 1 and under the two above assumptions $X$ follows a (shifted) Beta-Geometric distribution [52, 53], a compound distribution characterized by the two parameters $\alpha, \beta \in \mathbb{R}^+$, with the following PDF:

$$P(X = x|p) = \frac{B(\alpha + 1, \beta + x - 1)}{B(\alpha, \beta)}, \ x \in \{1, 2, 3, \ldots\}$$

It should be noted that the Beta-Geometric distribution is not a mixture of two probability distributions (Beta and Geometric), but a coherent compound distribution that results from a geometric process with variable success probability.

Despite its simplicity, the Beta-Geometric distribution is very versatile. This distribution can exhibit a wide variety of tail behaviors, depending on the parameters of the underlying Beta distribution. If the underlying Beta distribution has low variance, it results in a behavior similar to a geometric random variable. Otherwise, the variability of probabilities of success alone can result in very heavy-tailed behavior.

### Maximum-Likelihood for a Beta-Geometric distribution

Given $n$ independent realizations $x_1, \ldots, x_n$ of a Beta-Geometric random variable $X$, defined by parameters $\alpha$ and $\beta$ as described in Section The Beta-Geometric Distribution, the likelihood function is given by the product of their individual PDF:

$$L = \prod_{i=1}^{n} \frac{B(\alpha + 1, \beta + x_i - 1)}{B(\alpha, \beta)}$$

The corresponding log-likelihood is thus given by:

$$Log(L) = Log(B(\alpha + 1, \beta + x_i - 1)) - nLog(B(\alpha, \beta))$$

The maximum likelihood estimates for $\alpha$ and $\beta$, respectively $\hat{\alpha}$ and $\hat{\beta}$, can be obtained by numerically maximizing $Log(L)$ with respect to $\alpha$ and $\beta$ using statistical software such as the *betageometric* function of R package VGAM [54] or the *BGEPDF* of statistical software *data-plot* [55–57].

## Results and discussion

This section presents some properties of the empirical LOS in the four specialty departments which are direct consequences of the high frequency and statistical significance of extreme values. We subsequently propose a Beta-Geometric model of LOS and evaluates its goodness of fit.

### On the concentration of LOS

Heavy-tailed distributions tend to be dominated by a small percentage of observations. For LOS, this means that days of hospital would be concentrated among a small percentage of admissions or patients. Such a Power Law behavior which has been previously observed in the English healthcare system [58]. However, this effect cannot be adequately modeled by thin-tailed random variables, discarding outliers, or considering small samples.

It should be noted that this behavior is a natural consequence of the variability in patients' daily probabilities of discharge. An equal distribution of LOS is neither natural nor desirable. However, from a resource planning point of view, measuring this inequality is important in understanding and modeling healthcare systems in order to adequately price LOS. This is

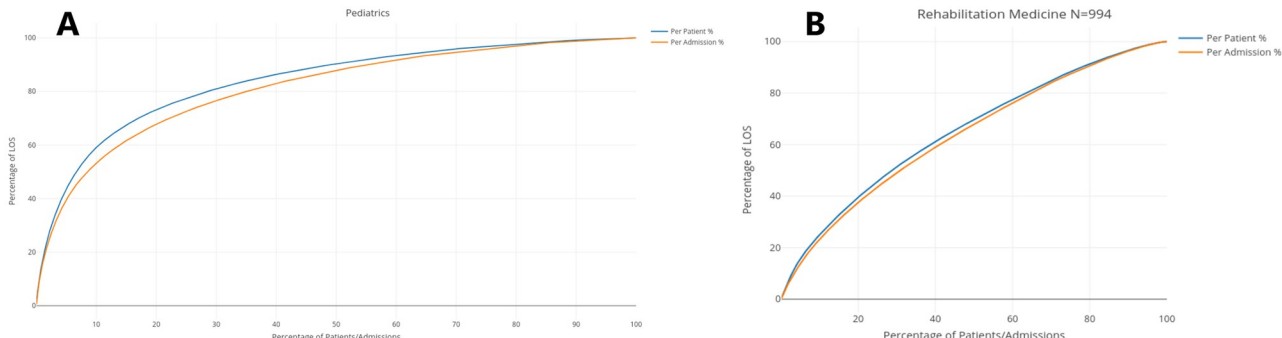

**Fig 2. Reversed Lorenz curve of LOS.** (A) Reversed Lorenz curve of LOS in Pediatrics. (B) Reversed Lorenz curve of LOS in Rehabilitation Medicine.

particularly true for high-demand hospitals operating close to capacity such as Siriraj hospital, where patients are directly competing for admission.

Fig 2A and 2B, S6A and S6B Fig represent the reversed Lorenz curves for LOS, by admission and by patient, in Pediatrics, Rehabilitation Medicine, Surgery, and OB/GYN respectively. These plots are Lorenz curves in which the horizontal axis is inverted. Thus, the horizontal axis represents the cumulative percentage of admissions/patients ordered by decreasing LOS, and the vertical axis represents the corresponding cumulative percentage of LOS consumed by that percentage of admissions/patients. These plots read for $x$% of patients/admissions have consumed $y$% of the LOS. In a thin-tailed distributions, though mildly concave, these plots would typically not show points of inflexion [59], from which the slope of the trendline markedly changes. These points, known as elbows, can serve as a basis for patient classification.

Table 3 presents the Gini index for the LOS per admission and per patient, for each specialty department, as well as the corresponding standard errors and 95% confidence intervals, estimated with bootstrap re-sampling [60].

As expected, we can observe that the heavier the tail of the distribution, the higher the inequality in the distribution of LOS, and the aggregation of LOS per patient tends to make the tails significantly heavier.

The distribution of LOS in Rehabiliation Medicine, once again, exhibits a behavior that is the closest to that of a thin-tailed random variable. However, we note an inflexion point at the left of the curve, where 3.05% of admissions concentrate 12.45% of LOS, and 3.6% of patients

**Table 3. Gini index for the LOS per admission and per patient, for each specialty department.**

| Department | Surgery | OB/GYN | Pediatrics | Reh. Med. |
|---|---|---|---|---|
| Nº of Admissions | 17949 | 19922 | 7499 | 994 |
| Gini index | .53 | .32 | .62 | .27 |
| Delta | 7.07 | 2.51 | 10.29 | 13.47 |
| Stand. error | 0.00633 | 0.00671 | 0.00880 | 0.03057 |
| Conf. interval | [0.517; 0.542] | [0.306; 0.333] | [0.602; 0.637] | [0.210; 0.329] |
| Nº of Patients | 14884 | 17483 | 5086 | 304 |
| Gini index | .55 | .32 | .67 | .30 |
| Delta | 8.78 | 2.84 | 16.44 | 48.59 |
| Stand. error | 0.00623 | 0.00671 | 0.00833 | 0.03029 |
| Conf. interval | [0.537; 0.562] | [0.306; 0.333] | [0.653; 0.686] | [0.240; 0.359] |

concentrate 12.36% of LOS. OB/GYN presents a similar but more extreme inflexion point, where 1.94% of admissions concentrate 10% of LOS and 2.58% of admissions concentrate 12.79%. Pediatrics and Surgery exhibit the most extreme discrepancies in the distribution of LOS between the top percentiles and the rest of the admissions/patients, thus resulting in lower bed turnover rates. In Pediatrics 6.7% of patients concentrate 50% of LOS, and 5.5% of admissions concentrate 40% of LOS, and in Surgery, 4.2% of admissions/patients concentrate 30% of LOS. The former specialty department also exhibits the widest gaps between the curves per admission and per patient, which illustrates the effect of multiple admissions on the concentration of LOS. The inflexion points determined by the percentile distribution of LOS allow for the definition of these thresholds in a way that takes actual resource consumption in a care unit into account.

## On revenue

As seen in Section On the concentration of LOS, the top percentiles of patients/admissions, in terms of LOS, can represent a very significant proportion of total LOS. This concentration of LOS also has financial effects. In Fig 3A and 3B, S7A and S7B Fig we have computed the average daily charge per admissions, as a function of LOS (i.e. revenue divided by LOS, for each discrete value of LOS), in Surgery, Rehabilitation Medicine, Pediatrics, and OB/GYN, respectively.

For the three departments of Surgery, OB/GYN and Pediatrics, we find that *ceteris paribus*, the mean revenue per day of an admission significantly decreases with the increase of LOS. That is up to a certain threshold, from which long LOS patients are approximately charged the same amount per day. Rehabilitation Medicine is once again a singular specialty department in this regards. Indeed, we find that the charge per day tends to increase with the increase of LOS for short stay patients, up to a point (from which long LOS could be defined) where it stabilizes for additional days of stays. The curves in Fig 3A and 3B can be accurately described as piecewise linear. Using the Elbow method [59], the respective thresholds for the pseudo-linearity of the charge per day function in each of the four specialty departments are given in Table 4.

In Fig 3A, as well as S7A and S7B Fig we observe a steeply declining, pseudo-linear curve up to a certain threshold of LOS, which could be considered the threshold for short LOS in terms of charge. For this group of patients, additional days of LOS result in a decrease in the charge per day (in addition to the opportunity cost resulting from the inability to admit

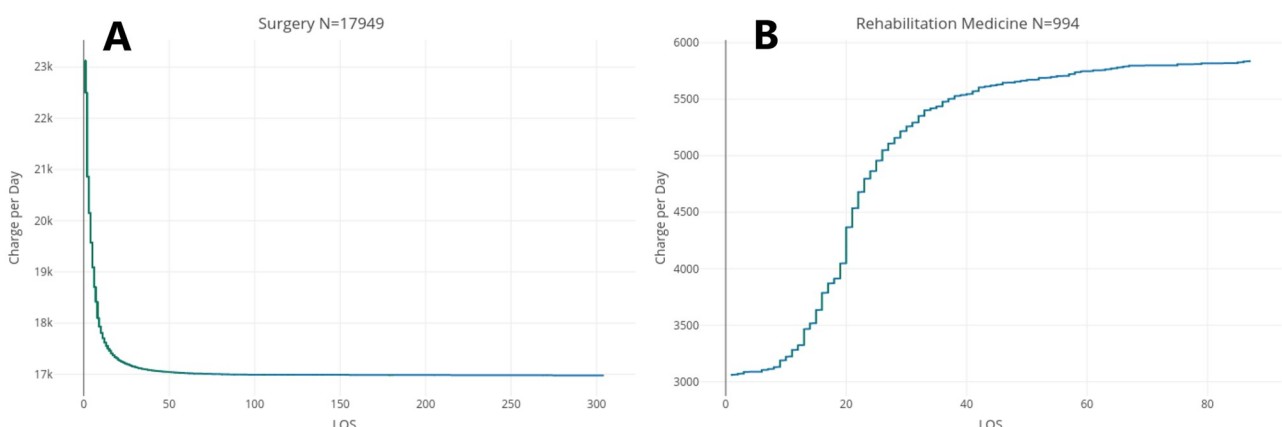

**Fig 3. Average charge per day.** (A) Average charge per day in Surgery. (B) Average charge per day in Rehabilitation Medicine.

**Table 4. Thresholds and piecewise correlation coefficients of LOS with average charge per day.**

| Department | Surgery | OB/GYN | Pediatrics | Reh. Med. |
|---|---|---|---|---|
| **1st Range** | 1 to 14 days | 1 to 11 days | 1 to 13 days | 1 to 9 days |
| Correlation | −.94 | −.88 | −.91 | +.83 |
| Stand. error | 0.00255 | 0.00337 | 0.00465 | 0.01771 |
| Conf. interval | [-0.945; -0.935] | [-0.887; -0.873] | [-0.919; -0.901] | [+0.795; +0.865] |
| **2nd Range** | 15 to 30 days | 12 to 17 days | 14 to 50 days | 10 to 39 days |
| Correlation | −.90 | −.90 | −.91 | +.73 |
| Stand. error | 0.00325 | 0.00309 | 0.00465 | 0.02170 |
| Conf. interval | [-0.906; -0.894] | [-0.906; -0.894] | [-0.919; -0.901] | [+0.687; +0.773] |
| **3rd Range** | 31 to 304 days | 18 to 119 days | 51 to 272 days | 40 to 87 days |
| Correlation | −.95 | −.94 | −.97 | +.90 |
| Stand. error | 0.00333 | 0.00242 | 0.00273 | 0.01384 |
| Conf. interval | [-0.955; -0.945] | [-0.945; -0.935] | [-0.975; -0.965] | [+0.873; +0.927] |

patients with a shorter stay and thus higher charge). Once the function the charge per day function reaches its minimum, it stabilizes for all higher LOS values.

Table 4 presents the piecewise Pearson correlations between LOS and average daily charge. The corresponding standard errors and 95% confidence intervals are also presented in this table. We find a very high negative piecewise linear correlation ($\leq -.88$) for each pseudo-linear range of the average charge per day function in Surgery, Ob/GYN, and Pediatrics, as detailed in Table 4. This hints at the fact that long LOS is not adequately priced, possibly because long LOS observations are discarded as outliers in analysis models.

These results highlight the important opportunity cost of long stays. Not only are long stay patients charged less per day, on average, but the difference in charge with shorter stay patients is compounded over a longer number of days. The significantly lower charge per day for longer stays is explained by the fact that the payment system in Thailand largely determines the charge based on DRGs, co-morbidities, and procedures performed, independently of LOS.

These observations are line with ones made in a large urban academic hospital of the state of Maryland, USA [22], where "outliers" (the quotes are intentionally so written in the referenced publication) have been found to be have an increasing impact on resource consumption, and it is recognized that the "the disproportionate costs of a few markedly high cost hospitalizations would not be fairly compensated under the DRG model".

This should not be seen as a call to make admission decisions based on revenue. It is however an invitation to consider a revision to the payment system to adequately price the opportunity cost resulting from long LOS. Siriraj's Rehabilitation Medicine department, however, seems to adequately price this opportunity cost (possibly because long LOS is expected in this department) as illustrated in Fig 3B.

The average charge per day increases with LOS, and shows a similarly high positive piecewise linear correlation when broken down into three pseudo-linear ranges. The average charge function in this specialty department is sigmoidal. We note that a change in the nature of the function from convex to concave occurs precisely at the 20 days mark. This is a reflection of the mixed nature of the distribution of LOS in Rehabilitation Medicine, with a marked difference in the distribution of values of LOS below and above 20 days.

Another question concerns the association between LOS, charge, and the discharge status of patients (broken down into positive and negative statuses, as described in Section Data Description and Basic Statistical Properties). Fig 4A, S8A and S8C Fig show

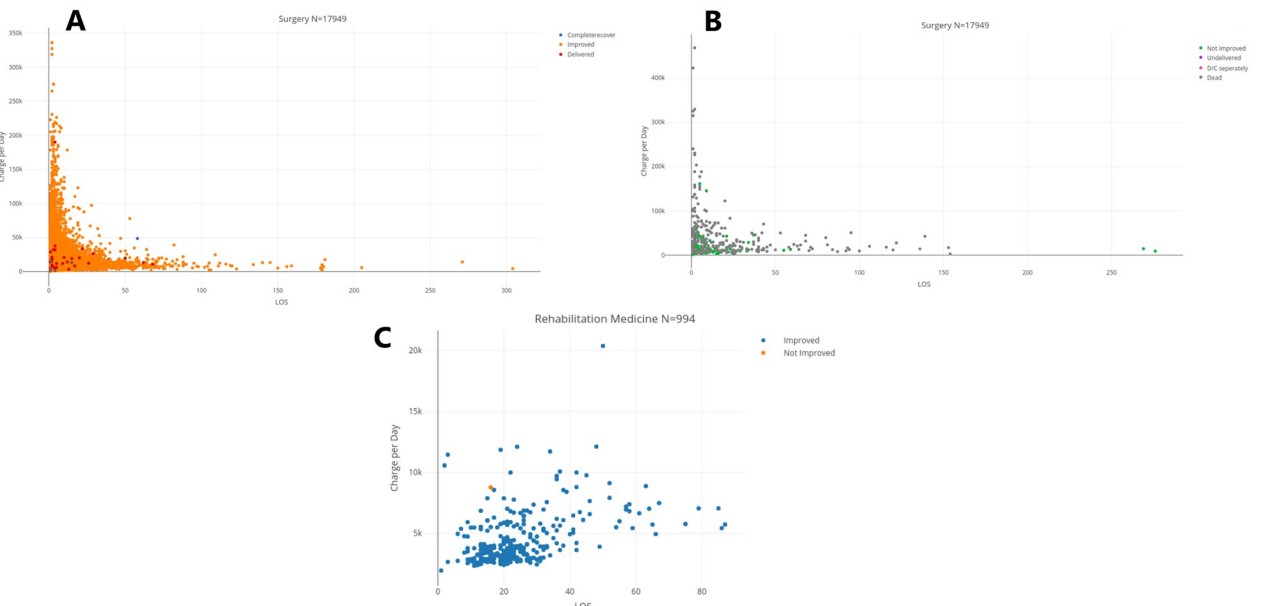

**Fig 4. Charge per day.** (A) Charge per day for positive discharge status in Surgery. (B) Charge per day for negative discharge status in Surgery. (C) Charge per day in Rehabilitation Medicine.

individual LOS and charges per day of admissions concluded with a positive discharge status in Surgery, OB/GYN, and Pediatrics, respectively, while Fig 4B, S8B and S8D Fig represents these variables for admissions concluded with a negative discharge status for the same respective departments. The Rehabilitation Medicine department having seen only a single admission with a negative discharge status ("Not improved"), the same data is represented in Fig 4C and is excluded from the following remark. Although discharges with a negative discharge status are much less frequent, we observe a similar distribution of LOS in each of the positive and negative discharge status group, as with the distribution of LOS over both groups.

Moreover, the distribution of LOS over each group shows a similar correlation with the average charge per day as the distribution over the whole. Overall, we can conclude that LOS, at this level of analysis (whole medical specialty departments) does not show any striking association with the discharge status. It may, however, possibly be the case of a more granular analysis per DRG, which is outside the scope of the present study.

## On the convergence of moments

Because of the preponderance of extreme values, one of the main concerns when suspecting subexponentiality in the study of random variables concerns the existence of moments, as described in Section Maximum to Sum Ratios. Indeed, the non-convergence of moments, or their very slow convergence (which requires extremely large sample sizes), renders the estimation of indicators such as the mean, variance, skewness or kurtosis from empirical observations impractical. Fig 5A and 5B represent the Maximum to Sum plot for LOS per admission and per patient, respectively, in Surgery.S9A, S9B, S9C and S9D Fig, as well as Fig 5C and 5D represent similar pairs of Maximum to Sum plots for OB/GYN, Pediatrics, and Rehabilitation Medicine. We note the convergence of first moments (mean) in all specialty departments. However,

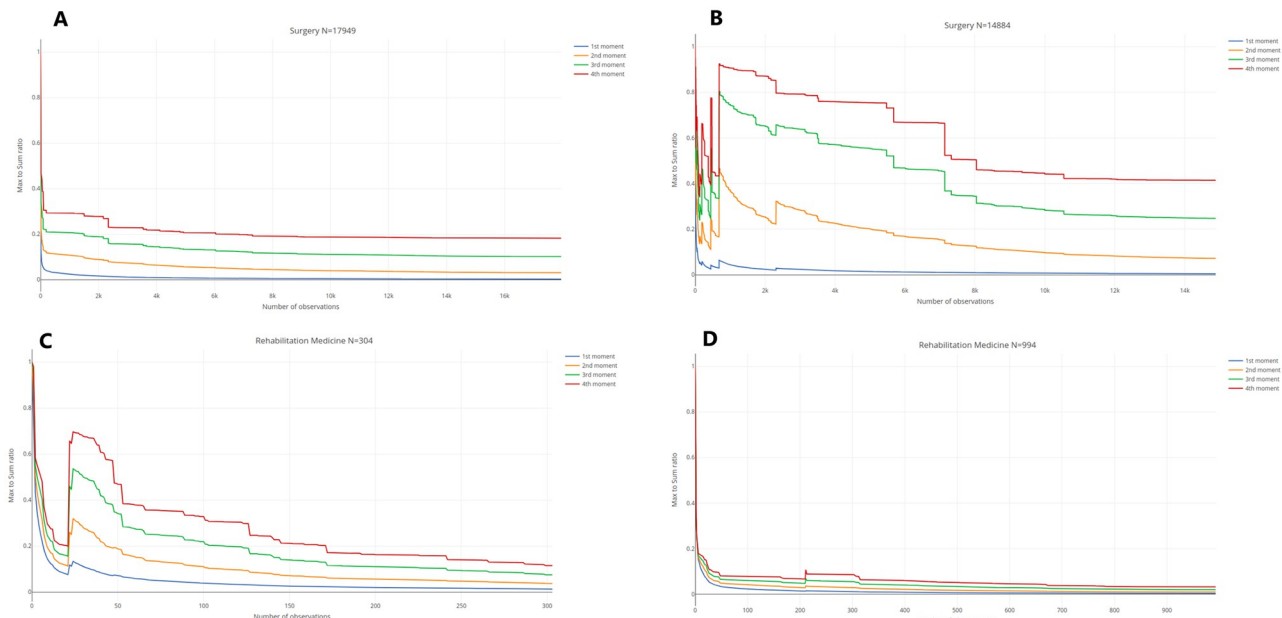

**Fig 5. Maximum to Sum ratios of LOS.** (A) Maximum to Sum ratios of LOS per admission in Surgery. (B) Maximum to Sum ratios of LOS per patient in Surgery. (C) Maximum to Sum ratios of LOS per admission in Rehabilitation Medicine. (D) Maximum to Sum ratios of LOS per patient in Rehabilitation Medicine.

second moments (variance), though apparently convergent, show a slow convergence even for our large datasets of tens of thousands of patients, which makes empirical estimates unreliable. Similarly, third and fourth moments (skewness and kurtosis respectively) cannot be reliably estimated from empirical data, except in the Rehabilitation Medicine department with per admission data.

This problem is particularly acute for LOS data per patient in Surgery and Pediatrics. As previously noted, the aggregation of LOS per patient makes the tail of the distributions of LOS heavier and the convergence of moments accordingly slower.

The novel information from these plots is the visible heavy tailedness of the distribution of LOS per patient in Rehabilitation Medicine.

We have excluded a Generalized Pareto distribution for each specialty department using the Pareto test included in R Package *ptsuite* [61, 62]. The resulting p-values were respectively $4.52 \cdot 10^{-193}$, 0, $1.24 \cdot 10^{-291}$, and 0 in Surgery, OB/GYN, Pediatrics, and Rehabilitation Medicine and are not consistent with generalized Pareto distributions. However, the Maximum to Sum plots are consistent with subexponential distributions, within which the Beta-Geometric is the best candidate because of its discreteness and consistency of assumptions in modeling LOS, as stated in Section The Beta-Geometric Distribution. This is further confirmed with QQ-plots and Mean Excess plots.

Indeed, in Fig 6A–6D, representing the QQ-plots of LOS per admission and per patient, in Surgery and Rehabilitation Medicine, we can observe a pseudo-linear behavior (including in Rehabilitation Medicine, past a certain threshold) that is consistent with a subexponential distribution. However, the marked difference between the head and tail of the QQ-plots in Fig 6C and 6D indicates that the distribution of LOS in Rehabilitation Medicine is possibly of a mixed nature, which is further confirmed by the Mean Excess plots in the next Section.

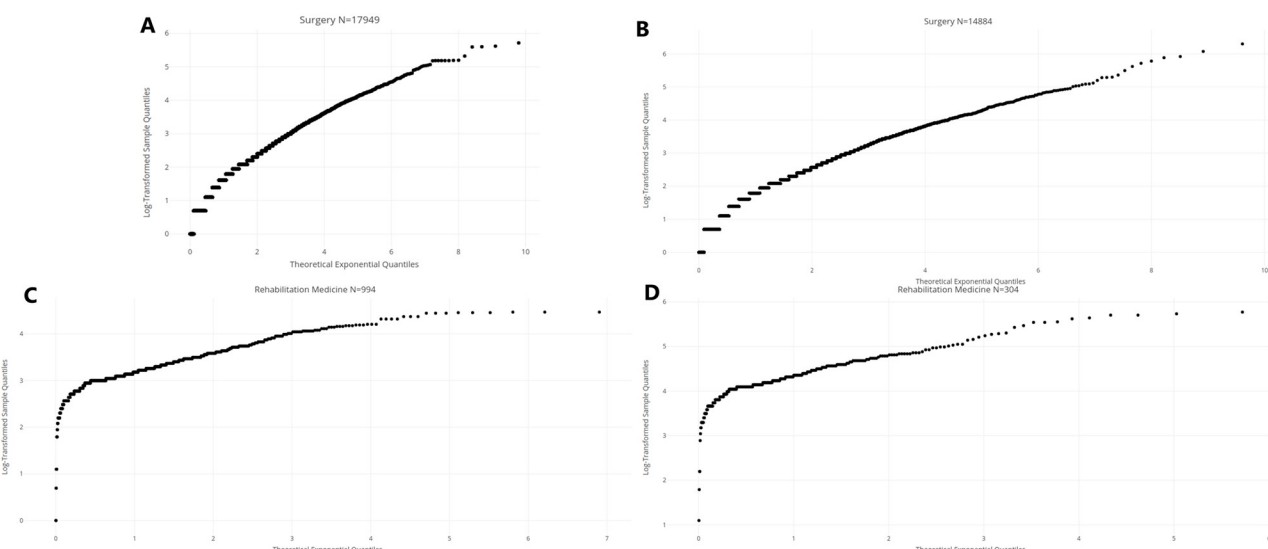

**Fig 6. Q-Q plot of LOS.** (A) Q-Q plot of LOS per admission in Surgery. (B) Q-Q plot of LOS per patient in Surgery. (C) Q-Q plot of LOS per admission in Rehabilitation Medicine. (D) Q-Q plot of LOS per patient in Rehabilitation Medicine.

## On expected residual length of stay

An increasing expected residual LOS is possibly the most important property that cannot be captured by thin-tailed models. As explained in Section Mean Excess Functions, this property can be understood as the expectation of LOS increasing the longer a patient has been in the system, or prosaically, the longer a patient has been admitted, the further they get from being discharged in subsequent days.

Fig 7A to 7D represent the mean excess plots for LOS per admission and per patient, in Surgery and Rehabilitation Medicine. They confirm the subexponential nature of these variables. Note that the decrease in the Mean Excess function at the rightmost points of the curve is a common effect known as the "finite sample bias" [44], wherein points close to the sample maximum would not be expected to further increase simply because no higher values have been observed yet.

The Mean Excess function of LOS in Rehabilitation Medicine, which is represented per admission in Fig 7C, and per patient in Fig 7D, once again stands out. The Mean Excess decreases for thresholds below 20. We see a dramatic change in the monotonicity of the Mean Excess Function, which goes from decreasing to increasing precisely at a value of 20 days. This indicates a heavy-tailed distribution for stays above 20 days.

Lastly, we observe that all increasing Mean Excess functions exhibit more linear behavior when LOS is aggregated by patient. This confirms, once again, the increase in the heaviness of tails that this aggregation induces.

## Modeling length of stay

Based on the evidence for a Beta-Geometric distribution of LOS in Pediatrics, Surgery, and OB/GYN, we have computed the maximum likelihood estimates of the $\alpha$ and $\beta$ parameters for LOS in these departments. As for the Rehabiliation Medicine department, and given the marked difference in the distribution before and after 20 days, and the evidence for a mixture distribution, we fit a Poisson distribution for values of LOS from 1 to 19 days with a Poisson distribution, whereas the distribution of values above 20 days is modeled with a shifted Beta-

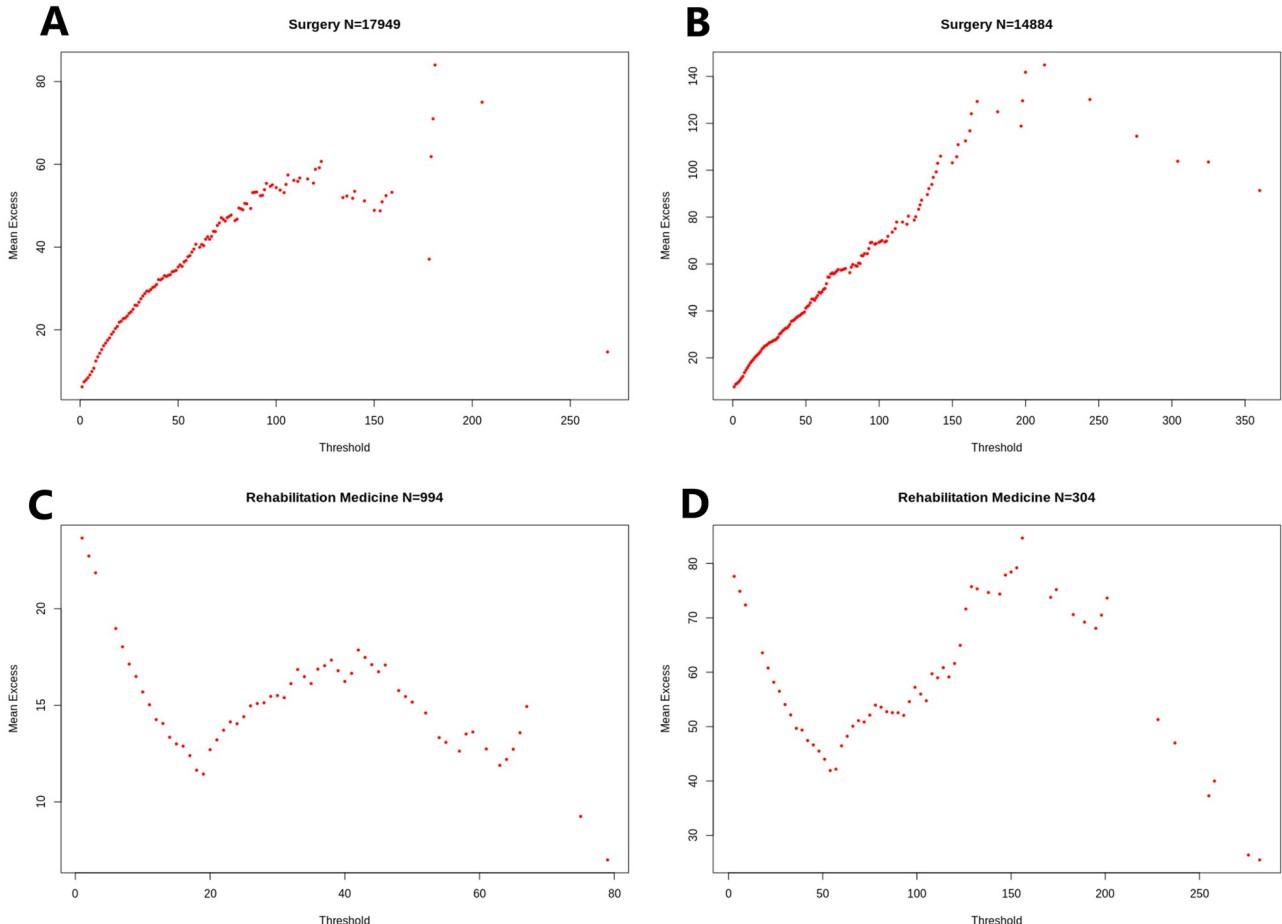

**Fig 7. Mean Excess function of LOS.** (A) Mean Excess function of LOS per admission in Surgery. (B) Mean Excess function of LOS per patient in Surgery. (C) Mean Excess function of LOS per admission in Rehabilitation Medicine. (D) Mean Excess function of LOS per patient in Rehabilitation Medicine.

Geometric. We subtract 20 days from all observations, fit a Beta-Geometric, then add 20 days to the resulting Beta-Geometric distribution.

We have generated a $10^5$ Monte-Carlo sample for the fitted distribution to the LOS in each of the four departments, both per admission and per patient. To evaluate goodness of fit, we use a discrete two-sample Kolmogorov–Smirnov goodness of fit test [63–65].

The Kolmogorov–Smirnov statistic (the $D$-statistic) more precisely quantifies the distance between the observed distribution function of the sample and the cumulative distribution function of the fitted distribution. We present the $\alpha$ and $\beta$ parameters found for the fitted Beta-Geometric distributions of LOS in the four departments, as well as the corresponding Kolmogorov–Smirnov distance, at 5% level of significance, in Table 5. For the Rehabilitation Medicine department this distribution only concerns values of LOS above 20 days as previously stated.

Moreover, Fig 8A and 8B respectively compare the histograms of the fitted and observed LOS per admission in Pediatrics and Rehabilitation medicine. Fig 8C and 8D respectively illustrate the comparison of cumulative distributions functions per patient in Surgery and per

**Table 5. Results of the Kolmogorov-Smirnov tests for fitted Beta-Geometric models with 95% confidence level.** A Poisson model was additionally used for stays below 20 days in Rehabilitation Medicine.

| Model | Surgery | OB/GYN | Pediatrics | Reh. Med. |
|---|---|---|---|---|
| **Admissions** | 17949 | 19922 | 7499 | 994 |
| $\alpha$ | 7.027677 | 7.1307 | 3.2695 | 37.3032 |
| $\beta$ | 38.58531 | 9.3826 | 17.6431 | 387.4253 |
| $D$-statistic | 0.0480 | 0.0398 | 0.0474 | 0.0605 |
| **Patients** | 14884 | 17483 | 5086 | 304 |
| $\alpha$ | 6.1900 | 13.9223 | 2.3282 | 10.4402 |
| $\beta$ | 39.6485 | 4.0856 | 14.9977 | 24.5310 |
| $D$-statistic | 0.0590 | 0.0483 | 0.0518 | 0.0632 |

admission in Rehabilitation Medicine, and Fig 9A and 9B the comparison of Mean Excess Functions for the distribution of LOS per patient in Surgery and per admission in Rehabilitation Medicine.

Fitting mixture distributions is a relatively under-studied problem in statistics, and overwhelmingly concerned with Gaussian mixtures [66, 67]. To the best of our knowledge, there haven't been any works regarding fitting a Poisson/Beta-Geometric mixture. As a result, fitting a theoretical distribution to the LOS in Rehabilitation medicine proved more challenging because of the mixed nature of the distribution of LOS in this department resulting from the artificial inflation of the frequency of value 20 days. However, for the three departments whose LOS only sees unconstrained variations (Surgery, Pediatrics, and OB/GYN), we obtain a satisfying fit with Beta-Geometric models and a 95% confidence level, as described in Table 5.

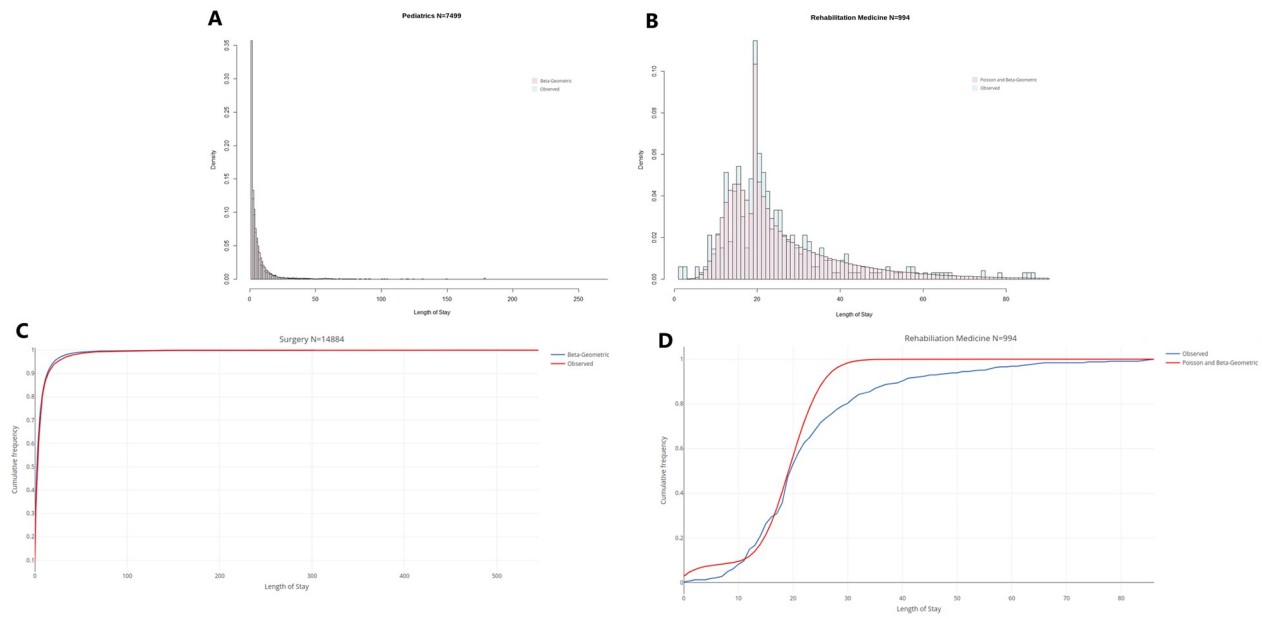

**Fig 8. Histogram and CDF of observed and fitted LOS.** (A) Histogram of observed and fitted LOS per admission in Pediatrics. (B) Histogram of observed and fitted LOS per admission in Rehabilitation Medicine. (C) CDF of observed and fitted LOS per patient in Surgery. (D) CDF of observed and fitted LOS per admission in Rehabilitation Medicine.

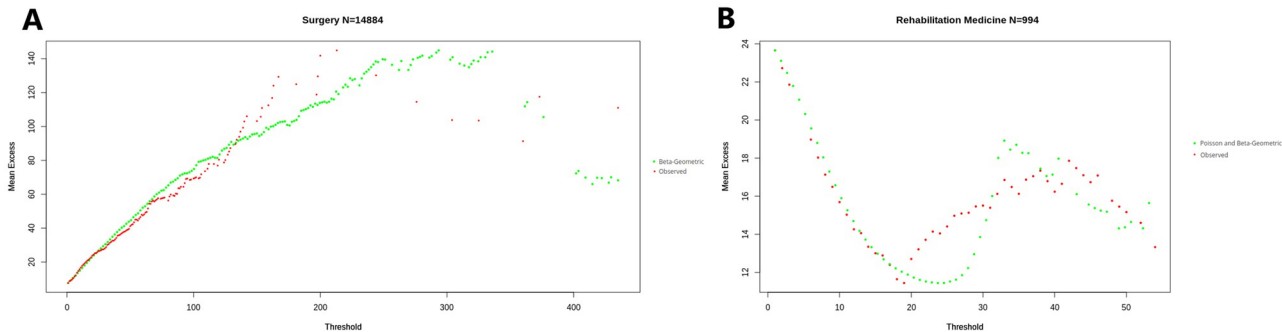

**Fig 9. Mean Excess function of observed and fitted LOS.** (A) Mean Excess function of observed and fitted LOS per patient in Surgery. (B) Mean Excess function of observed and fitted LOS per admission in Rehabilitation Medicine.

## Managerial implications

Section Results and Discussion identified and analyzed some empirical properties of LOS that have important implications for operational and financial planning at the level of healthcare units.

We have shown, in Section Gini Index and Lorenz Curves, that extreme value can result in large discrepancies in the distributions of LOS among admissions/patients. As discussed in Section Treatment of outliers, and throughout this paper, an important limitation in modeling LOS by DRG, and aggregating these models for the purpose of capacity planning, is that outliers (very long stays) appear less likely and less significant, when these outliers (which are only considered as such because of the thin-tailed nature of the models) have in fact a disproportionate impact on resource consumption.

Moreover, a key property of LOS studied in Section On Expected Residual Length of Stay is an increasing conditional expectation, as represented by the shape of the mean excess functions in Fig 7A to 7D. This is the case for all departments, including the Rehabilitation Medicine department past a certain threshold, as seen in Fig 7C and 7D. This property makes Length of Stay Lindy [48]; the longer a patient has stayed, the longer he/she would be expected to remain, with a corresponding increase in the expected opportunity cost of high LOS.

Thus, from the standpoints of both equity of access to healthcare and adequately reflecting the value offered to patients, the opportunity cost of long stays should be reflected in their pricing. Section On revenue showed that this opportunity cost is currently not adequately taken into account at Siriraj hospital.

Further, in Section On the Convergence of Moments, we have highlighted some practical difficulties in estimating the moments of LOS from empirical observations due to the slow convergence of the Law of Large Numbers. This *catastrophe principle*, inherent to subexponential random variables, makes models for forecasting precise values of LOS (e.g. least square regressions based on second moments) inapplicable. Lastly, we have illustrated the importance of reducing unnecessary readmissions and avoiding multiple admissions. As they compound to make the tails heavier, they exaggerate the previous challenges.

Based on the simple assumptions stated in The Beta-Geometric Distribution, we have shown, in Section Modeling Length of Stay, that seeing hospital admissions as a simple Geometric process with heterogeneous (Beta) probabilities of discharge adequately reproduces these properties and provides a good fit with observed LOS.

An actionable quantitative focus for capacity planning lies in accurately estimating the parameters of the Beta-Geometric process that governs a given LOS. This type of analysis has

the additional advantage of being scalable to different levels of decision-making (i.e. variations in the distribution of the discharge probability would result in different Beta-Geometric models of LOS in a hospital, departments, units, and within those for different DRG, type of patients, types of interventions, etc.). Further, the various graphical tools we have proposed could be employed to dynamically determine thresholds and use them for long-stay patient reviews. Moreover, pricing negotiations with payers should not only include the direct cost of treatment but also the opportunity cost of the expected LOS increasing in the tail, as well as the opportunity cost of being unable to admit patients with potentially shorter stays, as a result of tail events.

## Conclusions and limitations

This study showed that long stays (the tail of LOS distributions) have important consequences for resource consumption and revenue management in healthcare facilities and are not to be discarded as outliers. It is thus important that healthcare providers move away from simplistic, thin-tailed models and the disproportionate focus on linear regression, to align their quantitative models with those of payers (Extreme Value Theory is, after all, the mathematics of insurance). We have proposed a Beta-Geometric model for LOS that adequately reproduces these properties and shows a satisfying fit with empirically observed LOS in 46, 364 electronic health records, covering four specialty departments. An added advantage of the proposed model is that it offers a consistent model for capacity planning in a medical specialty department that can also be scaled down to individual DRGs, or up to the hospital as whole, by a simple adjustment of the parameters of the underlying Beta distribution of discharge probabilities. Moreover, the discreteness and simplicity of the assumptions that this model rests on offer an advantage over mixture distribution models. By Occam's razor, these models are not justified when a single (compound) probability distribution can adequately fit. However, we have found that manual restrictions on LOS (such as the preponderance of LOS value 20 in Rehabiliation Medicine) can produce empirical LOS distributions that are best described by mixture models. In these cases, modeling with a Beta-Geometric/Poisson mixture distribution proved more challenging and produced a lower-quality fit. A more refined analysis of LOS in Rehabilitation Medicine, over larger datasets than the one considered in our study (994 admissions), and with additional methodological developments on fitting mixed distributions would be potentially valuable.

## Supporting information

**S1 Fig. Monte Carlo illustrations of classical Extreme Value Theory results: Maximum to Sum Ratios.** (A) Maximum to Sum ratios for a Gaussian. (B) Maximum to Sum ratios for an Exponential. (C) Maximum to Sum ratios for a Beta-Geometric with $\alpha = 1.3$ and $\beta = 3$. (D) Maximum to Sum ratios for a Beta-Geometric with $\alpha = 1$ and $\beta = 3$.
(TIF)

**S2 Fig. Monte Carlo illustrations of classical Extreme Value Theory results: Mean Excess Functions.** (A) Mean Excess Functions for a Gaussian and Exponential. (B) Mean Excess Functions for a Pareto and Sub-Exponential. In S2A Fig, we plot the Mean Excess function of a truncated Gaussian (blue line) of mean 10 and variance 3 and Exponential (red line) of rate.1. Note that the chaotic perturbations at the extremity of the red curve are just the effect of the finite sample bias, i.e. the fact that points for very high order statistics in the plot are the result of very few observations [44]. S2B Fig present the mean excess functions of the exponential of the previous Gaussian, i.e. a Log-normal distribution (green line), as well as a Pareto

distribution of shape parameter.2. The axis labels have been voluntarily omitted to fit both curves within one plot and highlight the monotonic shape of the functions rather than specific values.
(TIF)

**S3 Fig. Monte Carlo illustrations of classical Extreme Value Theory results: Beta-Geometric Distribution.** (A) Mean Excess Functions for Beta-Geometric distributions. (B) Q-Q plot for a Beta-Geometric. We have generated Monte Carlo samples of size $10^6$ of Beta-Geometric random variables. As can be seen in S1D Fig, which represents the Maximum to Sum ratios for a Beta-Geometric with $\alpha = 1$ and $\beta = 3$ the mean and higher moments do not converge. S3A Fig presents the Mean Excess function for a Beta-Geometric with $\alpha = 1$ and $\beta = 3$ (blue points), $\alpha = 3$ and $\beta = 1$ (red points), and $\alpha = .5$ and $\beta = 1$ (green points). We can observe, the strictly increasing nature of these functions. Moreover, a log-transformed Pareto random variable is notoriously exponentially distributed. Hence a comparison of the theoretical quantiles of an Exponential random variable with those of the log-transform of empirical data is the basis of a visual Pareto test, known as the QQ (Quantile-Quantile) plot, Cf. Section, Gini Index and Lorenz Curves of [68]. A linear pattern indicates that the empirical data belongs to a generalized Pareto distribution and confirms heavy tails. S3B Fig presents a QQ plot for a $10^6$ Monte Carlo sample of a Beta-Geometric with $\alpha = 1$ and $\beta = 3$, in which linearity can be observed. In addition to the previous remarks concerning the non-convergence of moments and the increasing Mean Excess function, this observation makes us conclude that the Beta-Geometric can exhibit the behavior of a generalized Pareto model [69].
(TIF)

**S4 Fig. Monte Carlo illustrations of classical Extreme Value Theory results.** Beta-Geometric Distribution: The empirical limit of the ratio $\frac{1-F^{(2)}(x)}{1-F(x)}$ for $\alpha = 1.1$ and $\beta \in \{3, 5, 7, 9\}$. Though the focus of this paper does not lie in the theoretical study of random variables, it can be empirically verified that the tail of this random variable is at least heavy enough to belong to the sub-exponential class for some values of $\alpha$ and $\beta$ that result in even thinner tails than the values that seem adequate to model LOS in the dataset at hand. We have generated two random samples of $10^6$ observations each from two identical Beta-Geometric distributions with $\alpha \geq 1.1$ and $\beta \geq \alpha$ and verified the property defining subexponentiality.
(TIF)

**S5 Fig. Results for the omitted specialty departments; Histogram of LOS.** (A) Histogram of LOS per admission in Surgery. (B) Histogram of LOS per admission in Obstetrics and Gynaecology. (C) Histogram of LOS per patient in Surgery. (D) Histogram of LOS per patient in Obstetrics and Gynaecology.
(TIF)

**S6 Fig. Results for the omitted specialty departments; Reversed Lorenz curve of LOS.** (A) Reversed Lorenz curve of LOS in Surgery. (B) Reversed Lorenz curve of LOS in Obstetrics and Gynaecology.
(TIF)

**S7 Fig. Results for the omitted specialty departments; Average charge per day.** (A) Average charge per day in Pediatrics. (B) Average charge per day in Obstetrics and Gynaecology.
(TIF)

**S8 Fig. Results for the omitted specialty departments; Charge per day.** (A) Charge per day for positive discharge status in Pediatrics. (B) Charge per day for negative discharge status in

Pediatrics. (C) Charge per day for positive discharge status in Obstetrics and Gynaecology. (D) Charge per day for negative discharge status in Obstetrics and Gynaecology.
(TIF)

**S9 Fig. Results for the omitted specialty departments; Maximum to Sum ratios of LOS.** (A) Maximum to Sum ratios of LOS per admission in Pediatrics. (B) Maximum to Sum ratios of LOS per patient in Pediatrics. (C) Maximum to Sum ratios of LOS per admission in Obstetrics and Gynaecology. (D) Maximum to Sum ratios of LOS per patient in Obstetrics and Gynaecology.
(TIF)

**S10 Fig. Results for the omitted specialty departments; Q-Q plot of LOS.** ((A) Q-Q plot of LOS per admission in Pediatrics. (B) Q-Q plot of LOS per patient in Pediatrics. (C) Q-Q plot of LOS per admission in Obstetrics and Gynaecology. (D) Q-Q plot of LOS per patient in Obstetrics and Gynaecology.
(TIF)

**S11 Fig. Results for the omitted specialty departments; mean excess function of LOS.** (A) Mean Excess function of LOS per admission in Pediatrics. (B) Mean Excess function of LOS per patient in Pediatrics. (C) Mean Excess function of LOS per admission in Obstetrics and Gynaecology. (D) Mean Excess function of LOS per patient in Obstetrics and Gynaecology.
(TIF)

**S1 Data set.**
(XLSX)

# Acknowledgments

The first author is grateful to Prof. N. N. Taleb for his valuable advice and to Dr. Mark Simmerman for his careful proofreading of this manuscript.

# Author Contributions

**Conceptualization:** Nassim Dehouche.

**Data curation:** Nungruethai Torsuwan, Sakuna Taijan, Atthakorn Intharakosum.

**Formal analysis:** Nassim Dehouche.

**Investigation:** Nassim Dehouche.

**Methodology:** Nassim Dehouche.

**Project administration:** Sorawit Viravan.

**Resources:** Sorawit Viravan, Ubolrat Santawat, Nungruethai Torsuwan, Sakuna Taijan, Atthakorn Intharakosum, Yongyut Sirivatanauksorn.

**Software:** Nassim Dehouche.

**Supervision:** Yongyut Sirivatanauksorn.

**Validation:** Sorawit Viravan, Ubolrat Santawat, Yongyut Sirivatanauksorn.

**Visualization:** Nassim Dehouche.

**Writing – original draft:** Nassim Dehouche.

**Writing – review & editing:** Nassim Dehouche, Sorawit Viravan, Ubolrat Santawat, Yongyut Sirivatanauksorn.

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
