## [Decision Letter · Decision Letter 0]

10 May 2022

PONE-D-21-29331

Hospital Length of Stay: A cross-Specialty Analysis and Beta-Geometric Model.

PLOS ONE

Dear Dr. Viravan,

Thank you for submitting your manuscript to PLOS ONE. After careful consideration, we feel that it has merit but does not fully meet PLOS ONE’s publication criteria as it currently stands. Therefore, we invite you to submit a revised version of the manuscript that addresses the points raised during the review process.

Your manuscript has been assessed by an expert reviewer, whose comments are appended below. The reviewer raises important concerns about the study rationale, and the implementation and interpretation of the analysis. Ensure you address these points carefully in your revised manuscript.

Please note that we have only been able to secure a single reviewer to assess your manuscript. We are issuing a decision on your manuscript at this point to prevent further delays in the evaluation of your manuscript. Please be aware that the editor who handles your revised manuscript might find it necessary to invite additional reviewers to assess this work once the revised manuscript is submitted. However, we will aim to proceed on the basis of this single review if possible.

We look forward to receiving your revised manuscript.

Kind regards,

Joseph Donlan

Editorial Office

PLOS ONE

Journal Requirements:

2. In ethics statement in the manuscript and in the online submission form, please provide additional information about the patient records used in your retrospective study. Specifically, please ensure that you have discussed whether all data were fully anonymized before you accessed them and/or whether the IRB or ethics committee waived the requirement for informed consent. If patients provided informed written consent to have data from their medical records used in research, please include this information.

5. Please ensure that you refer to Figures 13, 19, and 20 in your text as, if accepted, production will need this reference to link the reader to the figure.

Reviewers' comments:

Reviewer's Responses to Questions

**Comments to the Author**

1. Is the manuscript technically sound, and do the data support the conclusions?

Reviewer #1: Partly

2. Has the statistical analysis been performed appropriately and rigorously? 

Reviewer #1: Yes

3. Have the authors made all data underlying the findings in their manuscript fully available?

Reviewer #1: Yes

4. Is the manuscript presented in an intelligible fashion and written in standard English?

Reviewer #1: Yes

5. Review Comments to the Author

Reviewer #1: #1)

The authors appropriately refer to individual DRG (line 134) and to decision-making (line 116). I think that the authors need to list several specific decisions that the authors are considering, not analyses per se but literally decisions. Consider which ones do not involve analyses by DRG and therefore are addressed by the current paper, because most of the decisions that I know about would be by DRG (e.g., guiding individual patients) or by knapsacks/ collections of DRGs (e.g., comparing hospitals conditional on the relative distributions of different DRGs). This matters because the authors currently are not performing analyses by DRG. If most decisions would be by DRG, then the authors’ analyses should be by DRG. The authors write at line 704 “it offers a consistent model for decision-support”. Please provide those decisions. At line 706 “can also be scaled down to individual DRGs.” Without being scaled down, for what decisions, and if needs to be scaled down, I would think that confirm results and usefulness when scaled down.

1a. For example, from line 119 “if the goal of the analysis … is hospital comparison, uniform trimming of LOS across all hospitals might be inappropriate.” My concern is that “might” is vague and the authors are not providing specific analyses of specific decisions. Under many conditions trimming works. See, for example, the following paper and its references: “Proportions of Surgical Patients Discharged Home the Same or the Next Day Are Sufficient Data to Assess Cases' Contributions to Hospital Occupancy,” Cureus 2021 doi: 10.7759/cureus.13826. Surely the authors’ modeling focused on long LOS is different than that paper. However, without the authors considering several specific decisions, the reliability of the authors’ claims seem limited.

1b. For example, the authors’ references to charges seem very likely to be dependent on individual payment systems (e.g., Line 507 “up to a certain threshold … patients are approximately charged the same amount per day”). Normally I would think that a suitable approach would be to have analyses of charges in supplemental content or a separate section specific to the authors’ country. However, with the authors not referring to specific decisions, I am unable to make a specific recommendation. The authors need please to be modeling a few (e.g., 2 or 3) specific decisions that are common and necessary.

1c. For example, “second moments (variance) … show a slow convergence” (line 569). Provide please specific common, important, decision pooling among DRGs for which one would want to estimate the variance. Referring to my first point, hospitals differ in their relative distributions of DRGs, and yet the authors have not modeled by DRG. Therefore, what is the decision? From line 574 “this problem is particularly acute for LOS data per patient in surgery.” Why would one be doing this, to decide or judge what?

#2)

Table 3 Gini index please add some standard errors, confidence intervals, equivalent to show that the indices are estimated sufficiently precisely given the sample sizes such that comparisons among them are appropriate. The same applies to the correlations in Table 4. Assure that discussion refers only to differences that are reliably (significantly) different correlations.

#3)

MINOR issues of writing

P1 abstract “would, counter-intuitively”. I do not follow why this is counter-intuitive. Consider deleting the phrase and just stating the behavior. Same line 599.

Line 383 “notoriously”, I do not follow what is the matter with the behavior. Why is this word needed? Consider deleting.

6. PLOS authors have the option to publish the peer review history of their article (what does this mean?). If published, this will include your full peer review and any attached files.

Reviewer #1: No

---

## [Author Response · Author response to Decision Letter 0]

21 Jul 2022

Thank you for allowing a resubmission of our manuscript, with an opportunity to address reviewers comments.

We are uploading (a) our point-by-point response to the review comments (below) (response to reviewers), (b) an updated manuscript with track changes, and (c) a clean updated manuscript without track changes.

Thank you, we have updated the manuscript and file names accordingly.

2. In ethics statement in the manuscript and in the online submission form, please provide additional information about the patient records used in your retrospective study. Specifically, please ensure that you have discussed whether all data were fully anonymized before you accessed them and/or whether the IRB or ethics committee waived the requirement for informed consent. If patients provided informed written consent to have data from their medical records used in research, please include this information.

The ethics statement has been updated and a statement about IRB approval and the anonymization of patient data has also been made in Section 5. Methods.

A dataset recording lengths of stay in the four medical specialty departments considered will be uploaded as supplemental data with this submission.

The ethics statement has been updated and a statement about IRB approval and the anonymization of patient data has also been made in Section 5. Methods.

5. Please ensure that you refer to Figures 13, 19, and 20 in your text as, if accepted, production will need this reference to link the reader to the figure.

Thank you. All figures in the Appendix have been referred to in the body of the revised manuscript.

Captions have been added to supporting information in the revised manuscript.

#1) The authors appropriately refer to individual DRG (line 134) and to decision-making (line 116). I think that the authors need to list several specific decisions that the authors are considering, not analyses per se but literally decisions. Consider which ones do not involve analyses by DRG and therefore are addressed by the current paper, because most of the decisions that I know about would be by DRG (e.g., guiding individual patients) or by knapsacks/ collections of DRGs (e.g., comparing hospitals conditional on the relative distributions of different DRGs). This matters because the authors currently are not performing analyses by DRG. If most decisions would be by DRG, then the authors’ analyses should be by DRG.

Thank you very much for raising this important point, allowing us to more precisely delimit the decision-making scope of the proposed statistical model. The brief answer to the above question is Capacity Planning; that is operational decisions about resources that are shared across multiple DRGs (hospital beds, vehicles, equipment, non-specialist workforce).

A recent (April 2022) literature review on Capacity Planning [1] lists the main ones of these decisions, as follows: 

● How many ward beds should be allocated for different specialties?

● How to align the hospital configuration to forecasted patients volumes?

● What proportion of time should be allocated to different specialties in operating theaters?

We agree that general mentions of “decision-making” in the initial manuscript were too broad and may be interpreted as including clinical decision-making.

Following this comment, Section 2. “Decision-Making Scope” has been added to the revised manuscript, and several additional references pertaining to capacity planning and the need for more refined models of LOS for operational decision-making have been compiled and referenced in this section. We have also revised the “Introduction” and “Managerial Implications” sections, and avoided the use of “decisions” and “decision-making” with more precise mentions of “capacity planning”.

Indeed, models of length of stay (LOS) by DRG are useful for clinical decision-making or quality/performance evaluation, as kindly raised by Reviewer #1 (guiding patients, quality/performance comparison), and mentioned in lines 58-64 of the initial manuscript. However, as discussed in the newly created “Decision-Making Scope” Section, making capacity decisions for shared resources requires simulating LOS volumes for all DRGs within a healthcare unit by fitting a theoretical distribution on empirical data (as is performed in the “Modeling Length of Stay” section), which also requires estimating the empirical moments of the distribution (mean, variance, skewness, kurtosis, etc.), among other input parameters.

This distinction between clinical decisions and capacity planning was in fact at the inception of our multidisciplinary team (covering clinical, surgery, operations, and finance specialties) and this paper. 

This work originated from our observation that statistical models of LOS by DRG, which are useful for guiding patients and making clinical decisions did not “scale up” to good models for capacity planning at Siriraj Hospital. As discussed in the paper, the main limitation in aggregating several (thin-tailed) statistical models by DRG is that outliers (very long stays) appear less likely and less significant. These outliers (which are only considered as such because of the thin-tailed nature of the models) have in fact a disproportionate impact on resource consumption (as the Section “On the concentration of LOS” shows).

Our observations have since been echoed by a recent publication that appeared since the submission of this manuscript. Indeed, [2] observes that “Outlier cases utilize a disproportionate and increasing share of hospital resources and available beds. The current tendency to exclude such outlier stays in data reporting due to assumed rare occurrence may need to be revisited”, and [3] note that “Neither are diagnosis-related groups an appropriate methodology for capacity planning …This means that they can be used for calculating prices but they say little about the mix of resources that is needed.”

The authors write at line 704 “it offers a consistent model for decision-support”. Please provide those decisions.

At line 706 “can also be scaled down to individual DRGs.” Without being scaled down, for what decisions, and if needs to be scaled down, I would think that confirm results and usefulness when scaled down.

Section 2. “Decision-Making Scope” of the revised manuscript hopefully clarifies these decisions. Moreover, we have replaced the general use of the term “decision-support” (which can indeed include clinical decision-making), by the more precise term “capacity planning”.

1a. For example, from line 119 “if the goal of the analysis … is hospital comparison, uniform trimming of LOS across all hospitals might be inappropriate.” My concern is that “might” is vague and the authors are not providing specific analyses of specific decisions. Under many conditions trimming works. See, for example, the following paper and its references: “Proportions of Surgical Patients Discharged Home the Same or the Next Day Are Sufficient Data to Assess Cases' Contributions to Hospital Occupancy,” Cureus 2021 doi: 10.7759/cureus.13826. Surely the authors’ modeling focused on long LOS is different than that paper. However, without the authors considering several specific decisions, the reliability of the authors’ claims seem limited.

Thank you, the reference you kindly provided offers an excellent illustration for the distinction between clinical decisions for individual cases and capacity planning at the level of a healthcare unit made in the revised manuscript. We have added this reference and contrasted it with the capacity planning models discussed in Section 2. “Decision-Making Scope”.

Please note that the decision in the reference kindly provided by reviewer #1 is clinical and concerns individual patients (“which cases may need to be postponed”). At an operational, larger scale, however, [4] observes that “outlier cases utilize a disproportionate and increasing share of hospital resources and available beds”. In our model, these are not outliers, but an essential component of the distribution (the tail) that determines many of its statistical properties.

1b. For example, the authors’ references to charges seem very likely to be dependent on individual payment systems (e.g., Line 507 “up to a certain threshold … patients are approximately charged the same amount per day”). Normally I would think that a suitable approach would be to have analyses of charges in supplemental content or a separate section specific to the authors’ country. However, with the authors not referring to specific decisions, I am unable to make a specific recommendation. The authors need please to be modeling a few (e.g., 2 or 3) specific decisions that are common and necessary.

Thank you for this comment. We have added details about the payment system in Thailand explaining the opportunity cost of long stays (payments are largely determined by DRGs, co-morbidities, and procedures performed, independently of LOS). We have also found these observations to be aligned with those made in the payment system of the USA (Medicare) by [4], and added a comparison in the same section. Please note that the purpose of the Section “On revenue” is to highlight the opportunity cost of long stays comparatively to shorter ones, which can be observed independently of the payment (although it is explained by it).

1c. For example, “second moments (variance) … show a slow convergence” (line 569). Provide please specific common, important, decision pooling among DRGs for which one would want to estimate the variance. Referring to my first point, hospitals differ in their relative distributions of DRGs, and yet the authors have not modeled by DRG. Therefore, what is the decision? From line 574 “this problem is particularly acute for LOS data per patient in surgery.” Why would one be doing this, to decide or judge what?

A simple usage of the variance (square of the standard deviation) is to measure the standard error for the mean LOS (given by ).

A slow converging variance/standard deviation means that it requires very large samples to be accurately measured.

Second moments (variance) is also the basis of least squares regression. As discussed in the “Managerial Implications”section” This catastrophe principle, inherent to subexponential random variables, makes models for forecasting precise values of LOS (e.g. least square

regressions based on second moments) inapplicable.”

Besides these, fitting distributions to empirical data for the purpose of capacity planning requires estimating the empirical moments of the variable at hand (mean, variance, skewness, kurtosis). The max to sum plots presented in the paper can be used to determine the appropriate sample sizes for the empirical moments to converge to their “true” theoretical value.

#2)

Table 3 Gini index please add some standard errors, confidence intervals, equivalent to show that the indices are estimated sufficiently precisely given the sample sizes such that comparisons among them are appropriate. The same applies to the correlations in Table 4. Assure that discussion refers only to differences that are reliably (significantly) different correlations.

Thank you. Standard errors and confidence intervals for both statistics have been computed and included in the revised manuscript, with a reference to the method used for the Gini coefficient (bootstrap re-sampling in R).

#3)

MINOR issues of writing

P1 abstract “would, counter-intuitively”. I do not follow why this is counter-intuitive. Consider deleting the phrase and just stating the behavior. Same line 599.

Thank you. We agree that the term subjectively assumes prior knowledge of the reader, and it has been removed in the revised manuscript. 

The intended meaning was that the observed behavior (“the longer a patient has been admitted, the higher their expectation of remaining admitted”, corresponding to the Lindy Effect https://en.wikipedia.org/wiki/Lindy_effect) goes against a layman’s expectation for the completion of a task (we expect something to complete sooner, the longer it has been going on), and what a thin-tailed model (Gaussian, Poisson, etc.) would produce.

Line 383 “notoriously”, I do not follow what is the matter with the behavior. Why is this word needed? Consider deleting.

Thank you. This term has been replaced with “is known to”, which was its intended meaning.

---

## [Decision Letter · Decision Letter 1]

31 Aug 2022

PONE-D-21-29331R1Hospital Length of Stay: A cross-Specialty Analysis and Beta-Geometric Model.PLOS ONE

Dear Dr. Viravan,

Thank you for submitting your manuscript to PLOS ONE. After careful consideration, we feel that it has merit but does not fully meet PLOS ONE’s publication criteria as it currently stands. Therefore, we invite you to submit a revised version of the manuscript that addresses the points raised during the review process.

Your manuscript has been assessed by an additional reviewer with expertise in epidemiological modelling, whose report can be found below. As you will see from the comments, the reviewer has requested that your manuscript be reorganized to comply with the STROBE guidelines for observational studies. Please consult the published guidelines (for example at https://doi.org/10.1371/journal.pmed.0040296) and revise your manuscript accordingly. Please also complete a copy of the relevant STROBE checklist (see https://www.strobe-statement.org/checklists/) and upload it as a supporting information file when you resubmit your manuscript.

We look forward to receiving your revised manuscript.

Kind regards,

Joseph Donlan

Senior Editor

PLOS ONE

Reviewers' comments:

Reviewer's Responses to Questions

**Comments to the Author**

1. If the authors have adequately addressed your comments raised in a previous round of review and you feel that this manuscript is now acceptable for publication, you may indicate that here to bypass the “Comments to the Author” section, enter your conflict of interest statement in the “Confidential to Editor” section, and submit your "Accept" recommendation.

Reviewer #2: (No Response)

2. Is the manuscript technically sound, and do the data support the conclusions?

Reviewer #2: Partly

3. Has the statistical analysis been performed appropriately and rigorously? 

Reviewer #2: Yes

4. Have the authors made all data underlying the findings in their manuscript fully available?

Reviewer #2: No

5. Is the manuscript presented in an intelligible fashion and written in standard English?

Reviewer #2: No

6. Review Comments to the Author

Reviewer #2: Dear author,

My main concern is related to the structure of the article. You did not follow the STROBE recommendations and the paper as it is very difficult to review.

Please look at the following article and try to follow a similar structure.

Shaaban, A. N., Peleteiro, B., & Martins, M. R. O. (2021). Statistical models for analyzing count data: predictors of length of stay among HIV patients in Portugal using a multilevel model. BMC Health Services Research, 21(1), 1-17.

7. PLOS authors have the option to publish the peer review history of their article (what does this mean?). If published, this will include your full peer review and any attached files.

Reviewer #2: **Yes: **Maria Rosario Oliveira Martins

---

## [Author Response · Author response to Decision Letter 1]

10 Jan 2023

Original Manuscript ID: PONE-D-21-29331

Original Article Title: “Hospital Length of Stay: A cross-Specialty Analysis and Beta-Geometric

Model”

To: PLOS One Editor

Re: Response to reviewers

Dear Editor,

We are grateful for the review comments received. The structure of the paper has been revised

according to the STROBE framework kindly recommended by Prof. Martins.

1. My main concern is related to the structure of the article. You did not follow the STROBE

recommendations and the paper as it is very difficult to review.

Please look at the following article and try to follow a similar structure.

Shaaban, A. N., Peleteiro, B., & Martins, M. R. O. (2021). Statistical models for analyzing count

data: predictors of length of stay among HIV patients in Portugal using a multilevel model. BMC

Health Services Research, 21(1), 1-17.

Thank you for this comment. We have restructured the manuscript accordingly.

2. Have the authors made all data underlying the findings in their manuscript fully available?

The PLOS Data policy requires authors to make all data underlying the findings described in

their manuscript fully available without restriction, with rare exception (please refer to the Data

Availability Statement in the manuscript PDF file). The data should be provided as part of the

manuscript or its supporting information, or deposited to a public repository. For example, in

addition to summary statistics, the data points behind means, medians and variance measures

should be available. If there are restrictions on publicly sharing data—e.g. participant privacy or

use of data from a third party—those must be specified.

Reviewer #2: No

Please kindly note that raw LOS data for the 46,364 considered admissions have been made

available in the file “LOS Data.xlsx”.

Kind regards,

The authors

---

## [Decision Letter · Decision Letter 2]

10 Apr 2023

PONE-D-21-29331R2Hospital Length of Stay: A cross-Specialty Analysis and Beta-Geometric Model.PLOS ONE

Dear Sorawit Viravan,

Thank you for submitting your manuscript to PLOS ONE. After careful consideration, we feel that your manuscript will likely be suitable for publication if it is revised to address the points below.   Therefore, my decision is "Minor Revision". We invite you to submit a revised version of the manuscript that addresses the points raised during the review process.

Please revise this paper.

We encourage you to submit your revision within forty-five days of the date of this decision.  If you will need more time than this to complete your revisions, please reply to this message or contact the journal office at plosone@plos.org. Please include the following items when submitting your revised manuscript:A rebuttal letter that responds to each point raised by the academic editor and reviewer(s). You should upload this letter as a separate file labeled 'Response to Reviewers'.A marked-up copy of your manuscript that highlights changes made to the original version. You should upload this as a separate file labeled 'Revised Manuscript with Track Changes'.An unmarked version of your revised paper without tracked changes. You should upload this as a separate file labeled 'Manuscript'.If applicable, we recommend that you deposit your laboratory protocols in protocols.io to enhance the reproducibility of your results. Protocols.io assigns your protocol its own identifier (DOI) so that it can be cited independently in the future. For instructions see: https://journals.plos.org/plosone/s/submission-guidelines#loc-laboratory-protocols. Additionally, PLOS ONE offers an option for publishing peer-reviewed Lab Protocol articles, which describe protocols hosted on protocols.io. Read more information on sharing protocols at https://plos.org/protocols?utm_medium=editorial-email&utm_source=authorletters&utm_campaign=protocols.

We look forward to receiving your revised manuscript.

Kind regards,

Oluwafemi Samson Balogun, Ph.D.

Academic Editor

PLOS ONE

Journal Requirements:

Reviewers' comments:

Reviewer's Responses to Questions

**Comments to the Author**

1. If the authors have adequately addressed your comments raised in a previous round of review and you feel that this manuscript is now acceptable for publication, you may indicate that here to bypass the “Comments to the Author” section, enter your conflict of interest statement in the “Confidential to Editor” section, and submit your "Accept" recommendation.

Reviewer #1: (No Response)

Reviewer #2: (No Response)

2. Is the manuscript technically sound, and do the data support the conclusions?

Reviewer #1: Partly

Reviewer #2: Partly

3. Has the statistical analysis been performed appropriately and rigorously? 

Reviewer #1: Yes

Reviewer #2: Yes

4. Have the authors made all data underlying the findings in their manuscript fully available?

Reviewer #1: Yes

Reviewer #2: No

5. Is the manuscript presented in an intelligible fashion and written in standard English?

Reviewer #1: Yes

Reviewer #2: No

6. Review Comments to the Author

Reviewer #1: I reviewed the authors’ original submission and was invited to review this R2. I do not think that the authors have addressed decision-making. The authors list two decisions on lines 41 and 42, “How to align the hospital configuration to forecasted patient volumes?” and “What proportion of time should be allocated to different specialties in operating theatres?”. I think that the authors need to refer to specific decisions, with specific mathematics, that has been studied in detail and show that there are PRECISE decisions to be made with the mathematics. This was not done in the R1. Otherwise, if the authors prefer, say that this is basic science. Please appreciate that basic science is fine. However, to point out how the authors are not being precise, line 42 refers to operating theatres, and yet there is no modeling for turnover times, lengths of surgical cases, etc. I am in no way recommending that be added, it would be out of scope. I use this example to highlight that the authors are not studying the allocation of operating room times. They have NOT done what they report was done (line 42) nor that I recommended.

In the R1, the authors added reference 21 for deciding about individual surgeries to be postponed. This is a specific decision, not conjecture. There was similarly another paper by the same authors on forecasting for individual hospitals by time series showing accuracy of the modeling (Cureus 2020;12:e10847). The topic does indeed differ from that of the authors. The authors explain in their section (added for R1) that the “scale of modeling” is different. This seems entirely reasonable assessment. I’m not recommending that the authors add more about this. But, use these examples that precise consideration of a decision at their higher “scale of modeling” means, please, being precise and showing that the dependent variables that they are including are sufficient for the decision.

I consider further the managerial implications. The statements lines 708-721 about LOS seem fully accurate. However, I do not appreciate that the authors have shown that their new mathematics was necessary for lines 708-721. In other words, the facts seem accurate, and they are managerial implications of LOS being skewed. However, please add PRECISE information about how SPECIFICALLY these implications are from the authors’ new results, as compared with knowledge about LOS beforehand.

Regarding pricing, the authors refer to revenue management. Please provide DETAILS of PRECISELY what decisions the authors are suggesting. The authors know LOS and charges in retrospect, so how is this revenue management? For revenue management, that’s prospective, before the product is purchased. The “revenue” in the authors’ context is to be obtained from whom and when? Furthermore, at least for surgery as I know, the models regarding hospital financial decision-making long-term incorporate other variables than LOS and “charges,” specifically the indirect/intangible value (“utility”) associated with individual types of patients. For example, suppose that there were low or negative contribution margin for vascular surgery because of long LOS outliers. That does not mean that one can remove vascular surgery, because without vascular surgeons, there can be no obstetrics, general surgery, urology, trauma, etc., at the hospital. Hospitals generally need all specialties because the patients and the complications involve multiple specialties. This is why without the authors referring to SPECIFIC decisions with DETAILS from other studies, I recommend that the authors markedly reduce the background sections, implications sections, etc., and simply present their (very) reasonable methods and results.

Reviewer #2: Dear authors,

In my opinion the paper is much more a statistical paper than a health sciences/ public health paper; the structure is still not adequate; please dont include all the mathematical equations (they are very usefull for statistitians, but not for the most of the public health lectors)

The authors must include less statistics methods and more public health discussion and more relevance on their approach to the public health field.

7. PLOS authors have the option to publish the peer review history of their article (what does this mean?). If published, this will include your full peer review and any attached files.

Reviewer #1: No

Reviewer #2: No

While revising your submission, please upload your figure files to the Preflight Analysis and Conversion Engine (PACE) digital diagnostic tool, https://pacev2.apexcovantage.com/. PACE helps ensure that figures meet PLOS requirements. To use PACE, you must first register as a user. Registration is free. Then, login and navigate to the UPLOAD tab, where you will find detailed instructions on how to use the tool. If you encounter any issues or have any questions when using PACE, please email PLOS at figures@plos.org. Please note that Supporting Information files do not need this step.<quillbot-extension-portal></quillbot-extension-portal>

---

## [Author Response · Author response to Decision Letter 2]

14 Jun 2023

Response to Reviewer #1’s Feedback

We are very grateful for your nuanced feedback and agree with your concerns regarding the

precision of decisions made based on our mathematical findings. We concur that the

manuscript’s main focus is on fundamental statistical properties, specifically relating to the

choice of the best fitting statistical model for LOS data, and less on managerial and clinical

extrapolations which have not been sufficiently supported.

Moreover, we emphasize that study does not only propose a model for decision-making - the

beta-geometric distribution discussed in Section "4.5. Modeling Length of Stay," but also offers a

detailed analysis of the inherent properties of empirical Length of Stay (LOS) using Extreme

Value Theory, a well-established area of statistics.

Reviewer #1: I reviewed the authors’ original submission and was invited to review this

R2. I do not think that the authors have addressed decision-making. The authors list two

decisions on lines 41 and 42, “How to align the hospital configuration to forecasted

patient volumes?” and “What proportion of time should be allocated to different

specialties in operating theatres?”. I think that the authors need to refer to specific

decisions, with specific mathematics, that has been studied in detail and show that there

are PRECISE decisions to be made with the mathematics. This was not done in the R1.

Otherwise, if the authors prefer, say that this is basic science. Please appreciate that

basic science is fine.

However, to point out how the authors are not being precise, line 42 refers to operating

theatres, and yet there is no modeling for turnover times, lengths of surgical cases, etc. I

am in no way recommending that be added, it would be out of scope. I use this example

to highlight that the authors are not studying the allocation of operating room times.

They have NOT done what they report was done (line 42) nor that I recommended.

We appreciate your highlighting the need to clarify the primary focus of our research and its

broader context within the field. Strictly speaking, the main decision addressed by our study is

indeed mathematical and is: "What is the most appropriate random variable to model hospital

length of stay?" We have highlighted that this is an active area of research with our comparative

literature review, summarized in Table 1. of the manuscript.

However, as you astutely point out, decision-making related to capacity planning involves other

critical factors beyond LOS, including additional operational and clinical data that are not directly

considered by our model. These decisions are also highly contingent on the specific resources,

strategies, and constraints of individual healthcare facilities.

In response, in the revised “Decision-Making Scale” section, we emphasize more clearly that

our research focuses primarily on the mathematical modeling of LOS, while acknowledging the

broader, multi-dimensional context within which this modeling occurs. With this revision, we

aimed to clarify that our model serves as one piece of the larger puzzle of capacity management

decision-making, rather than a comprehensive solution. Moreover, we limit our conclusions,

throughout the paper, to the modeling scale that is supported by our empirical data (i.e.

specialty departments).

In the R1, the authors added reference 21 for deciding about individual surgeries to be

postponed. This is a specific decision, not conjecture. There was similarly another paper

by the same authors on forecasting for individual hospitals by time series showing

accuracy of the modeling (Cureus 2020;12:e10847). The topic does indeed differ from that

of the authors. The authors explain in their section (added for R1) that the “scale of

modeling” is different. This seems entirely reasonable assessment. I’m not

recommending that the authors add more about this. But, use these examples that

precise consideration of a decision at their higher “scale of modeling” means, please,

being precise and showing that the dependent variables that they are including are

sufficient for the decision.

In response to these observations, in the Section “Scope of Decision-Making”, we have

highlighted the work of Pearson et al. and Devapriya et al. with StratBAM, as examples of

decision-making within the scope of our model. These studies utilized LOS models to estimate

the need for resources, such as Intensive Care Beds, across various DRGs. That being said, we

have also emphasized that statistical models for LOS, such as the one we propose, represent

just one component of hospital capacity planning software. Such planning also demands the

consideration of other critical factors. Thus, we additionally define the scope of our work with

less ambiguity with “It is this specific aspect - the decision of the statistical model to represent

LOS - that our present paper primarily addresses.”

Further, the various graphical tools we have proposed could be employed to dynamically

determine thresholds and use them for long-stay patient reviews.

I consider further the managerial implications. The statements lines 708-721 about LOS

seem fully accurate. However, I do not appreciate that the authors have shown that their

new mathematics was necessary for lines 708-721. In other words, the facts seem

accurate, and they are managerial implications of LOS being skewed. However, please

add PRECISE information about how SPECIFICALLY these implications are from the

authors’ new results, as compared with knowledge about LOS beforehand.

We emphasize that study does not only propose a model for decision-making - the

beta-geometric distribution discussed in Section "4.5. Modeling Length of Stay," but also offers a

detailed analysis of the inherent properties of empirical Length of Stay (LOS) using Extreme

Value Theory, a well-established area of statistics. Our study is, for instance, in line with Baek,

H., Cho, M., Kim, s., Hwang, H., Song, M., Yoo, S. Analysis of length of hospital stay using

electronic health records: A statistical and data mining approach. PLoS One, 13(4), 2018 in

studying the fundamental statistical properties of LOS, on a different scale, and Bonetti, M.,

Cirillo, P., Musile Tanzi, P., Trinchero, E. An Analysis of the Number of Medical Malpractice

Claims and Their Amounts. PLoS ONE 11(4): e0153362, 2016.

doi:10.1371/journal.pone.0153362, utilizing Extreme Value Theory.

We have abridged the “Managerial Implications” section and revised it to underscore, not only

the obvious skewness of LOS, but the heavy-tailedness of its distribution. In practical terms, this

implies that outlier observations, often viewed as surprising or exceptional in the clinical context,

are in fact relatively common and constitute a significant portion of the overall distribution. This

is a crucial finding that addresses observations made by clinicians and in studies such as

Hughes et al., 2021, where prolonged stays are perceived as exceptional.

In essence, our contribution seeks to recalibrate these intuitive expectations, which are often

based on the assumptions of a 'Normal' distribution, to reflect the actual statistical properties of

LOS.

Regarding pricing, the authors refer to revenue management. Please provide DETAILS of

PRECISELY what decisions the authors are suggesting. The authors know LOS and

charges in retrospect, so how is this revenue management? For revenue management,

that’s prospective, before the product is purchased. The “revenue” in the authors’

context is to be obtained from whom and when? Furthermore, at least for surgery as I

know, the models regarding hospital financial decision-making long-term incorporate

other variables than LOS and “charges,” specifically the indirect/intangible value

(“utility”) associated with individual types of patients. For example, suppose that there

were low or negative contribution margin for vascular surgery because of long LOS

outliers. That does not mean that one can remove vascular surgery, because without

vascular surgeons, there can be no obstetrics, general surgery, urology, trauma, etc., at

the hospital. Hospitals generally need all specialties because the patients and the

complications involve multiple specialties. This is why without the authors referring to

SPECIFIC decisions with DETAILS from other studies, I recommend that the authors

markedly reduce the background sections, implications sections, etc., and simply

present their (very) reasonable methods and results.

We appreciate your constructive feedback and agree with your points regarding the complexity

of hospital revenue management. Given this complexity, we've revised our manuscript to focus

solely on our main contributions, namely, identifying properties of length of stay (LOS) using

Extreme Value Theory, and proposing the beta-geometric distribution for modeling LOS.

In response to your comments, we've withdrawn certain conjectures related to revenue

management that weren't directly supported by our research. We have kept our discussion

focused on our key findings based on empirical data from Siriraj Hospital (i.e. the observation

that long stays are currently underpriced and their opportunity cost compounds).

Thank you for your valuable insights that have helped us refine our study and improve the

precision of our presentation.

Reviewer #2: Dear authors,

In my opinion the paper is much more a statistical paper than a health sciences/ public

health paper; the structure is still not adequate; please dont include all the mathematical

equations (they are very usefull for statistitians, but not for the most of the public health

lectors)

The authors must include less statistics methods and more public health discussion and

more relevance on their approach to the public health field.

We appreciate your feedback and understand your perspective. However, our paper does not

claim to have relevance for public health. It is fundamentally rooted in Operations Management

and focuses on the mathematical properties of Length of Stay (LOS) as a random variable. As

such, it is primarily intended for operations researchers and data science within healthcare.

For these reasons, the keywords associated with our manuscript are “Length of Stay, Statistical

Modeling, Extreme Value Theory, Beta-Geometric Distribution.”

Recognizing your concerns, we have revised our manuscript to clearly state this scope in the

Decision-making section, helping to clarify the audience and context for our work. Furthermore,

we've made an effort to limit conjectural statements about broader health implications and to

focus more on our core results, in order to avoid any confusion regarding the scope and intent

of our paper.

We trust that these changes align with your insightful comments and hope that they help clarify

the nature and intent of our research.

---

## [Decision Letter · Decision Letter 3]

22 Jun 2023

Hospital Length of Stay: A cross-Specialty Analysis and Beta-Geometric Model.

PONE-D-21-29331R3

Dear Dr. Sorawit Viravan,

We’re pleased to inform you that your manuscript has been judged scientifically suitable for publication and will be formally accepted for publication once it meets all outstanding technical requirements.

Kind regards,

Oluwafemi Samson Balogun, Ph.D.

Academic Editor

PLOS ONE

Additional Editor Comments (optional):

Reviewers' comments:

Reviewer's Responses to Questions

**Comments to the Author**

1. If the authors have adequately addressed your comments raised in a previous round of review and you feel that this manuscript is now acceptable for publication, you may indicate that here to bypass the “Comments to the Author” section, enter your conflict of interest statement in the “Confidential to Editor” section, and submit your "Accept" recommendation.

Reviewer #1: All comments have been addressed

Reviewer #2: All comments have been addressed

2. Is the manuscript technically sound, and do the data support the conclusions?

Reviewer #1: Yes

Reviewer #2: Yes

3. Has the statistical analysis been performed appropriately and rigorously? 

Reviewer #1: Yes

Reviewer #2: Yes

4. Have the authors made all data underlying the findings in their manuscript fully available?

Reviewer #1: Yes

Reviewer #2: Yes

5. Is the manuscript presented in an intelligible fashion and written in standard English?

Reviewer #1: Yes

Reviewer #2: Yes

6. Review Comments to the Author

Reviewer #1: (No Response)

Reviewer #2: Dear athors

Thank you very much for the answers. I have no more questions

The paper in my opinion is in conditions to be published

7. PLOS authors have the option to publish the peer review history of their article (what does this mean?). If published, this will include your full peer review and any attached files.

Reviewer #1: No

Reviewer #2: No

<quillbot-extension-portal></quillbot-extension-portal>

---

## [Editor Report · Acceptance letter]

3 Jul 2023

PONE-D-21-29331R3 

Hospital Length of Stay: A cross-Specialty Analysis and Beta-Geometric Model 

Dear Dr. Viravan:

I'm pleased to inform you that your manuscript has been deemed suitable for publication in PLOS ONE. Congratulations! Your manuscript is now with our production department. 

Kind regards, 

on behalf of

Dr. Oluwafemi Samson Balogun 

Academic Editor

PLOS ONE